# Safety and Immunogenicity of Inactivated Whole Virion COVID-19 Vaccine CoviVac in Clinical Trials in 18–60 and 60+ Age Cohorts

**DOI:** 10.3390/v15091828

**Published:** 2023-08-29

**Authors:** Ilya V. Gordeychuk, Liubov I. Kozlovskaya, Aleksandra A. Siniugina, Nadezhda V. Yagovkina, Vladimir I. Kuzubov, Konstantin A. Zakharov, Viktor P. Volok, Maria S. Dodina, Larissa V. Gmyl, Natalya A. Korotina, Rostislav D. Theodorovich, Yulia I. Ulitina, Dmitry I. Vovk, Marina V. Alikova, Anna A. Kataeva, Anna V. Kalenskaya, Irina V. Solovjeva, Elena V. Tivanova, Larissa Y. Kondrasheva, Antonina A. Ploskireva, Vasiliy G. Akimkin, Ksenia A. Subbotina, Georgy M. Ignatyev, Anastasia K. Korduban, Elena Y. Shustova, Ekaterina O. Bayurova, Alla S. Zhitkevich, Daria V. Avdoshina, Anastasia N. Piniaeva, Anastasia A. Kovpak, Liliya P. Antonova, Yulia V. Rogova, Anna A. Shishova, Yury Y. Ivin, Svetlana E. Sotskova, Konstantin A. Chernov, Elena G. Ipatova, Ekaterina A. Korduban, Aydar A. Ishmukhametov

**Affiliations:** 1Chumakov Federal Scientific Center for Research and Development of Immune-and-Biological Products of Russian Academy of Sciences, 108819 Moscow, Russia; 2Institute for Translational Medicine and Biotechnology, Sechenov First Moscow State Medical University, 117418 Moscow, Russia; 3Kirov State Medical University of the Ministry of Health of Russia, 610998 Kirov, Russia; 4Healthcare Unit No. 163 of Federal Medical Biological Agency of Russia, 630559 Novosibirsk Region, Russia; 5Eco-Safety Scientific Research Center, 196143 Saint-Petersburg, Russia; 6Lomonosov Moscow State University, 119991 Moscow, Russia; 7RIC-Pharma, 123298 Moscow, Russia; 8Central Research Institute of Epidemiology of the Federal Service for Surveillance on Consumer Rights Protection and Human Wellbeing, 111123 Moscow, Russia; 9Perm State Medical University named after E. A. Wagner of the Ministry of Healthcare of the Russian Federation, 614000 Perm, Russia

**Keywords:** COVID-19, inactivated vaccine, neutralizing antibodies, phase I-II clinical trials, 18–60 age cohort, 60+ age cohort

## Abstract

We present the results of a randomized, double-blind, placebo-controlled, multi-center clinical trial phase I/II of the tolerability, safety, and immunogenicity of the inactivated whole virion concentrated purified coronavirus vaccine CoviVac in volunteers aged 18–60 and open multi-center comparative phase IIb clinical trial in volunteers aged 60 years and older. The safety of the vaccine was assessed in 400 volunteers in the 18–60 age cohort who received two doses of the vaccine (n = 300) or placebo (n = 100) and in 200 volunteers in 60+ age cohort all of whom received three doses of the vaccine. The studied vaccine has shown good tolerability and safety. No deaths, serious adverse events (AEs), or other significant AEs related to vaccination have been detected. The most common AE in vaccinated participants was pain at the injection site (*p* < 0.05). Immunogenicity assessment in stage 3 of Phase II was performed on 167 volunteers (122 vaccinated and 45 in Placebo Group) separately for the participants who were anti-SARS-CoV-2 nAB negative (69/122 in Vaccine Group and 28/45 in Placebo Group) or positive (53/122 in Vaccine Group and 17/45 in Placebo Group) at screening. On Day 42 after the 1st vaccination, the seroconversion rate in participants who were seronegative at screening was 86.9%, with the average geometric mean neutralizing antibody (nAB) titer of 1:20. A statistically significant (*p* < 0.05) increase in IFN-γ production by peptide-stimulated T-cells was observed at Days 14 and 21 after the 1st vaccination. In participants who were seropositive at screening but had nAB titers below 1:256, the rate of fourfold increase in nAB levels was 85.2%, while in the participants with nAB titers > 1:256, the rate of fourfold increase in nAB levels was below 45%; the participants who were seropositive at screening of the 2nd vaccination did not lead to a significant increase in nAB titers. In conclusion, inactivated vaccine CoviVac has shown good tolerability and safety, with over 85% NT seroconversion rates after complete vaccination course in participants who were seronegative at screening in both age groups: 18–60 and 60+. In participants who were seropositive at screening and had nAB titers below 1:256, a single vaccination led to a fourfold increase in nAB levels in 85.2% of cases. These findings indicate that CoviVac can be successfully used both for primary vaccination in a two-dose regimen and for booster vaccination as a single dose in individuals with reduced neutralizing antibody levels.

## 1. Introduction

The ongoing pandemic of coronavirus disease 2019 (COVID-19), which is caused by severe acute respiratory syndrome coronavirus 2 (SARS-CoV-2), has claimed more than 6.9 million lives as of May 2023 [1] In order to ensure the availability of vaccines against COVID-19 to every country in the world, all existing vaccine platforms and manufacturing capacities were employed, including genetic [2,3], vector [4,5], subunit [6] and inactivated [7,8,9,10] vaccines. Besides vaccination, some pre-exposure and post-exposure prophylaxis and treatment regimens are used, including direct-acting antivirals, specific monoclonal antibodies, anti-inflammatory drugs and immunomodulators [11].

While the registered COVID-19 vaccines demonstrate high immunogenicity and high short-term efficacy, it has been shown that post-vaccination antibody levels decrease over time [12,13], which has led to an implementation of annual and even semiannual booster vaccinations in unfavorable epidemiological conditions in several countries, including Russia, setting even greater demands for the safety profile of the vaccines used.

Despite generally lower post-vaccination neutralizing antibody (nAB) levels as compared to vector and mRNA vaccines, inactivated vaccines have been instrumental in the prevention of severe cases of COVID-19, with tens of billions of doses administered worldwide, showing exceptional safety [8,14]. Moreover, the reported adverse events associated with the use of genetic vaccines are the main concern for their use in the context of the need for periodic booster doses. During a pandemic, the high risks associated with increased rates of mortality and critical illness outweigh the risk of the relatively rare adverse events, but in the post-pandemic era, the use of these vaccines for routine immunization requires in-depth risk-benefit assessment [15]. Therefore, the safety profile of the inactivated vaccines makes them a promising tool for routine vaccination with frequent booster doses.

We developed a β-propiolactone-inactivated whole virion vaccine CoviVac, which has previously shown no signs of acute/chronic, reproductive, embryo- and fetotoxicity, or teratogenic effects in the antenatal and postnatal periods of development, as well as no allergenic properties in rodents and nonhuman primates in preclinical trials [16]. Here, we report the safety and immunogenicity of inactivated whole virion vaccine CoviVac in clinical trials in 18–60 and 60+ age cohorts.

## 2. Materials and Methods

### 2.1. Ethical Approval

The protocol of the CoviVac safety and immunogenicity assessment study in volunteers aged 18–60 years (VKI-I/II-08/20) and its supplementary documentation were approved by the Ethics Committee of the Ministry of Health of the Russian Federation (No. 502 from 21 September 2020). Additionally, the protocol was approved by the local Ethics Committees of the clinical sites, namely Kirov State Medical University of the Ministry of Health of Russia [No. 13/2020 from 28 September 2020]; Healthcare Unit No. 163 of FMBA of Russia, [No.1 from 11 October 2020]; Eco-Safety Scientific Research Center [No. 157 from 1 October 2020]. The study protocol was registered at clinicaltrials.gov (ID NCT05046548).

The protocol of the CoviVac safety and immunogenicity assessment study in volunteers aged 60+ years (VKI-P-II-07/21) and its supplementary documentation were approved by the Ethics Committee of the Ministry of Health of the Russian Federation (No. 288 from 28 September 2021). Additionally, the protocol was approved by the local Ethics Committees of the three clinical sites, namely Mechnikov Research Institute of Vaccines and Sera [No. 483/01 from 3 November 2021], Perm State Medical University [No. 10 from 27 October 2021] and Healthcare Unit No. 163 of FMBA of Russia [No.1 from 28 October 2021]. The study protocol was registered at clinicaltrials.gov (ID NCT05765773). The studies were conducted in accordance with the Declaration of Helsinki guidelines, ICH GCP and Russian regulations.

### 2.2. Study Design and Participants

#### 2.2.1. Design and Participants in 18–60 Age Cohort

A randomized, double-blind, placebo-controlled, multi-center clinical trial of the tolerability, safety, and immunogenicity of the inactivated whole virion concentrated purified coronavirus vaccine CoviVac was performed in adult volunteers aged 18–60 years.

Studies in 18–60 age cohort were performed in 3 consequent Stages: Stage 1 included the first 15 volunteers (10 inoculated with the vaccine and 5 with placebo twice with 14 days interval, on days 0 and 14); Stage 2 included 185 participants (140 inoculated with the vaccine and 45 with placebo twice with 14 days interval) and started after the post-vaccination observation period in Stage 1; Stage 3 included 200 participants (150 inoculated with the vaccine and 50 with placebo twice with 14 days interval) (Figure 1). Safety assessment was performed in Stages 1–3, and immunogenicity assessment was performed in Stage 3.

During initial screening, all participants in the 18–60 age cohort were tested for SARS-CoV-2 infection by PCR in nasopharyngeal swabs. The participants of Stages 1 and 2 were also screened at enrolment for anti-SARS-CoV-2 IgM and IgG antibodies by ELISA. Individuals positive in either test were not included. The participants of Stage 3 were included regardless of their SARS-CoV-2 IgM and IgG status at enrolment. Individuals with a history of SARS-CoV-1, SARS-CoV-2, or MERS infection, confirmed contact with COVID-19 patients, previous severe allergic reactions, tuberculosis, cancer, and mental or autoimmune diseases were also not included. All participants were screened for eligibility on the basis of their health status, including their medical history, vital signs, physical examination and laboratory test results, and were enrolled after providing signed and dated informed consent forms. Detailed study protocol can be found at clinicaltrials.gov (ID NCT05046548).

#### 2.2.2. Design and Participants in 60+ Age Cohort

Safety and immunogenicity assessment of CoviVac in the 60+ age cohort was performed in an open multi-center phase IIb clinical trial.

At initial screening, all participants in the 60+ age cohort were tested for SARS-CoV-2 infection via PCR using nasopharyngeal swabs. All participants were screened for eligibility on the basis of their health status, including their medical history, vital signs, physical examination and laboratory test results, and were enrolled after providing signed and dated informed consent forms.

The study included healthy volunteers and volunteers who had a history of chronic diseases with a stable course, aged 60 years and older at the time of inclusion, males and females, who provided signed and dated informed consent form, were able to comply with the requirements of the study protocol (i.e., filling out self-observation diaries and coming to scheduled visits) and did not meet the study non-inclusion criteria. The non-inclusion criteria were PCR or ELISA-confirmed SARS-CoV-2 infection in the previous 6 months, other acute respiratory diseases in the previous 4 weeks, hyperthermia at the time of vaccination, neutropenia, severe and/or uncontrolled cardiovascular, bronchopulmonary, neuroendocrine, gastrointestinal, liver, kidney, hematopoietic, immune system diseases, medical history of serious allergic reactions, serious post-vaccination reactions, Guillain–Barré syndrome, tuberculosis, cancer, autoimmune diseases immunosuppressive conditions, recent large blood donations (450 mL or more), splenectomy and anorexia.

Out of 230 screened volunteers, 200 were included, and 168 received all three doses of the vaccine and entered immunogenicity assessment 21 days after completing the vaccination course (Figure 2). Details of the inclusion, non-inclusion and exclusion criteria can be found in the study protocol (clinicaltrials.gov ID NCT05765773).

### 2.3. Randomization and Blinding

Randomization was performed using the sealed envelope method on the day of the 1st vaccination. Distribution of the participants into Vaccine or Placebo Groups in the 18–60 age cohort was carried out using random number generator software.

After assigning a three-digit randomization number to a participant, a sealed envelope with the randomization number was given to a healthcare worker who administered the vaccine. The participants in the 18–60 age cohort did not know which preparation they were receiving. After the vaccination, a separate physician observed the participant for possible AEs. The laboratory specialists were provided with the samples marked with coded numbers and thus were also blinded.

All study participants in the 60+ age cohort received randomization numbers used for the monitoring of possible adverse events (AEs) and immunogenicity assessment, but no further distribution between groups was performed. All participants in the 60+ age cohort received the vaccine.

### 2.4. Procedures

CoviVac vaccine (manufactured by the Chumakov FSC R&D IBP RAS as described previously [16]) is a whole virion β-propiolactone inactivated concentrated purified vaccine prepared in Vero cells and supplemented with aluminum hydroxide. The final vaccine preparation did not contain β-propiolactone, which was controlled using high-performance liquid chromatography assay during the late stages of the manufacturing process. Preparations were provided as single dose (0.5 mL) vials with sterile liquid that was injected intramuscularly into deltoid muscle. A placebo preparation contained buffer saline used in the vaccine preparation and the equivalent amount of aluminum hydroxide.

In the 18–60 age cohort, the vaccine was injected twice at 14-day intervals. Participants were observed for any AEs for 30 min following the administration of each dose, and then daily until Day 15–17 (1–3 days after the 2nd vaccination), with subsequent follow-up visits at Days 20 (only for Stages 1 and 2), 28–31 and 42–45 (for all Stages). Self-recording of AEs by participants was performed using diaries. During the visits, vital signs measurement and physical examination were performed. Neurological assessment was performed before each vaccination, 3 days after each vaccination and 28 days after the 2nd vaccination. Urine and blood samples were collected at screening and days 3, 7, 10, 14, 17, 20, 28 and 42 after the 1st vaccination for hematological and biochemical analysis and immunological tests (at Stage 3). ECG was carried out at screening and once on days 2–6 after the 1st vaccination.

In the 60+ age cohort, the vaccine was injected three times with 21 days intervals. Participants were observed at clinical sites for any AEs for at least 30 min following the administration of each dose with subsequent phone calls 5–8 h later on the same day, at scheduled appointments 2, 7 and 14 days after each vaccination and also 21 days after the 3rd vaccination. Reporting of AEs by participants was performed using self-observation diaries. At screening, on each day of vaccination and two days after each vaccination, and 21 days after the 3rd vaccination, vital signs measurement and physical examination were performed. Serum samples were obtained from all participants five days before the 1st vaccination and then on Day 42 (21 days after 2nd vaccination) and Day 63 (21 days after 3rd vaccination) for immunogenicity assessment. Nasopharyngeal swabs for SARS-CoV-2 PCR test were performed at screening, on the days of the 2nd and 3rd vaccination and 21 days after the 3rd vaccination. Urine and blood samples were collected at screening and days 2, 23, 42, 44 and 63 after the 1st vaccination for hematological and biochemical analysis.

### 2.5. Safety Assessment

AEs were monitored until Day 42–45 after the 1st vaccination (Day 28–31 after the last vaccination) for the 18–60 age cohort and until Day 63 (21 days after 3rd vaccination) for the 60+ age cohort. The solicited local AEs were pain, indurations, hematomas, swelling, itching and hypersensitivity at the injection site, and solicited systemic AEs were fever, fatigue or malaise, nervous system disorders (headache, lightheadedness), musculoskeletal and connective tissue disorders (arthralgia, myalgia), disorders of the respiratory system and mediastinal organs (pain/sore throat, nasal congestion, rhinorrhea, cough, shortness of breath), gastrointestinal tract disorders (nausea, vomiting, diarrhea, impaired appetite). All unsolicited AEs were reported by participants throughout the study.

AEs were graded as mild, moderate or severe according to the severity scoring chart (Appendix A) and assessed for the probability of their relation to the tested vaccine (highly probable/certain, probable, possible, unlikely, unrelated, unknown).

### 2.6. Immunogenicity Assessment

For serum separation, blood was collected into Vacuette tubes with clot activator and serum separator gel. Serum was separated via centrifugation, aliquoted into 1.7 mL tubes and stored frozen for further analysis for anti-SARS-CoV-2 antibodies.

Humoral immunity against SARS-CoV-2 was assessed in a virus neutralization test (NT). NT was performed using SARS-CoV-2 strain PIK35 in Vero cells as described previously [16]. Strain PIK35 and the CoviVac vaccine strain both belong to prototype B.1.1 SARS-CoV-2 PANGO lineage (Kozlovskaya et al., 2020). Serum samples that did not show neutralization in 1:8 dilution were considered negative.

Serum samples from randomly picked participants were additionally tested in ELISA for total antibodies against RBD with CoronaPass Total (Genetico, Russia) and for total antibodies to S protein trimer (Vector-Best, Russia).

Interferon-gamma (IFN-γ) production in response to stimulation with SARS-CoV-2 S-protein peptides was assessed using QuantiFERON SARS-CoV-2 Starter Set (Qiagen, Germany), using two SARS-CoV-2 S protein peptide pools (Ag1 and Ag2) designed to stimulate the production of IFN-γ by S-protein-specific CD4 (Ag1) and CD4/CD8 (Ag2) T cells, which was detected by ELISA according to the manufacturer’s protocol. A Mitogen tube served as a positive control. The baseline IFN-γ level in the Nil tube (unstimulated lymphocytes) was subtracted from the IFN-γ level in the antigen tubes and the Mitogen tube to correct for background or non-specific IFN-γ signal. The results were reported in international units of IFN-γ per milliliter of whole blood (IU/mL) using a dilution curve for the standard supplied with the kit. A calibration curve was generated at each assay run. Samples were considered reactive for an IFN-γ response if the IFN-γ levels obtained from the tubes with stimulus were 0.01 IU/mL above baseline levels obtained from the unstimulated control.

### 2.7. Study Outcomes

For the 18–60 age cohort, the primary outcomes for safety assessment were the frequency and severity of AEs within the observation period (72 h, 7 days after each vaccination and 28 days after the 2nd vaccination). The primary immunogenicity endpoints were the geometric mean nAB titer (GMT) and seroconversion rate (defined as a 4-fold or higher increase in GMT of specific antibodies in the NT) 28 days after the 2nd vaccination. Secondary endpoints included seroconversion rates at Days 7 and 14 and Months 2, 3, 4, 5 and 6 after the 2nd vaccination.

For the 60+ age cohort, the primary outcomes of immunogenicity assessment were the proportion of study participants with an increase in GMT of specific antibodies in the NT and seroconversion rate (defined as 4-fold or higher increase in GMT of specific antibodies in the NT) on Day 21 after the 3rd vaccination in comparison with the data obtained in healthy volunteers in the 18–60 age cohort. The primary outcomes of safety assessment were the frequency, type and severity of AEs and their relation to vaccination in comparison with the data obtained in healthy volunteers in the 18–60 age cohort.

### 2.8. Statistical Analysis

All parameters were evaluated using descriptive statistics (mean, standard deviation, percentile, medians, min and max).

For age cohort 18–60, the sample size was determined based on the requirements of the national guidelines, and the distribution of the participants between Vaccine Group and Placebo Group was 3:1.

Value comparisons (for example, GMT) were made using ANOVA or Mann–Whitney tests. *p* values below 0.05 (two-sided) were considered to be significant. Frequencies (%) were compared with Chi-square test or Fisher’s exact test. Analysis of variance or Friedman’s analysis was used for comparisons of the parameters over time. Spearman’s rank correlation coefficient was used for correlation analysis.

For age cohort 18–60, variance analysis with repeated measures was used to analyze changes in quantitative indices measured more than 2 times over time. For posterior comparisons, a paired Bonferroni-corrected *t*-test was used. The sphericity of the covariance–dispersion matrix was assessed using the Mauchly sphericity criterion. If the sphericity condition was not satisfied, corrections for the number of degrees of freedom Greenhouse-Geisser (if ε < 0.75) or Huench-Feldt (if ε ≥ 0.75) were applied. If the data did not meet the conditions of variance analysis, Friedman’s analysis was used for comparisons of quantitative measures over time. Dunn’s method with Bonferroni correction was used for a posteriori comparison. Confidence intervals for the proportions were calculated using the Wilson method. The Kolmogorov–Smirnov test was used to test for normality. For all used criteria, in addition to the level of significance (p), the values of the corresponding criterion statistics are indicated with the indication of degrees of freedom. For the analysis of variance, the sum of squares (SS) and mean square (MS) are also indicated.

## 3. Results

### 3.1. Demographic and Anthropometric Data

#### 3.1.1. 18–60 Age Cohort

The study profile in the 18–60 age cohort is presented in Figure 1. Overall, starting from 3 October 2020, 498 volunteers were screened for eligibility at three clinical sites. Based on clinical, laboratory and instrumental examinations, 400 volunteers aged 18–60 years were enrolled in the study. All enrolled participants were considered healthy and had no previous or current diagnosed comorbid conditions. All participants were randomized using the sealed envelope method into two groups: 300 participants were assigned to receive the vaccine, and 100 participants were assigned to receive a placebo (two injections, 0.5 mL, 14 day intervals for both the Vaccine and Placebo Groups).

Two out of three hundred participants from the Vaccine Group withdrew consent prior to the administration of the 1st dose. Thus, a safety assessment was performed on 398 (298 in the Vaccine Group and 100 in the Placebo Group) participants of the study. Participants who contracted SARS-CoV-2 infection during the observation period were not excluded from the safety assessment.

Out of 200 (150 in the Vaccine Group and 50 in the Placebo Group) participants initially included in Stage 3 for immunogenicity assessment, a total of 33 (28 from the Vaccine Group and 5 from the Placebo Group) were excluded during the observation period as they contracted SARS-CoV-2 infection or withdrew consent. Thus, an immunogenicity assessment was performed on 167 (122 in the Vaccine Group and 45 in the Placebo Group) participants of Stage 3.

The demographic characteristics of the participants in the safety and immunogenicity assessment populations were similar across treatment groups in terms of sex, age, and mean body mass index (BMI) (Table 1). The mean age of the study participants was 33.1 and 33.5 years, the female/male proportion was 36.9/63.1 and 34/66, and BMI was 23.8 and 24 kg/m^2^ for the Vaccine Group and the Placebo Group, correspondingly. The differences in demographic and anthropometric parameters between the Vaccine and Placebo Groups were statistically insignificant.

#### 3.1.2. 60+ Age Cohort

The study profile in the 60+ age cohort is presented in Figure 2. In total, 230 volunteers aged 60 years and older, healthy and having a history of chronic diseases with stable course, were screened for eligibility at three clinical sites. Based on inclusion and non-inclusion criteria, 200 volunteers were enrolled at screening and therefore included in the vaccine safety assessment. The demographic characteristics of the study participants in the 60+ age cohort are summarized in Table 2. The study participants in the 60+ age cohort had a mean age of 67.26 years, 130 (65%) were female and 70 (35%) were male, with a mean BMI of 27.09. This corresponds with the demographic distribution by sex in the Russian population in the 60+ age cohort: 62.6% female vs. 37.4% male [17].

In total, out of 200 participants that passed screening, 32 were excluded during the study: 16 (8%) had confirmed SARS-CoV-2 infection, 9 (4.5%) contracted other ARIs, 1 (0.5%) was in quarantine after contact with a person with confirmed COVID-19, 2 (1%) did not come to scheduled visits, 1 (0.5%) received drugs that were prohibited by the study protocol, and 2 (1%) withdrew consent. One participant was excluded on the day of the 1st vaccination before vaccine administration, as he had a medical history of massive blood loss, anemia (hemoglobin below 80 g/L), agranulocytosis, and neutropenia (below 1000 neutrophils per microliter) that was not detected at screening.

Thus, out of 200 participants who were initially included, 199 (99.5%) received the first dose of the vaccine, 175 (87.5%) received the second dose and entered immunogenicity assessment after two doses, and 168 (84%) received the third dose and entered immunogenicity assessment after three doses according to the protocol.

### 3.2. Tolerance, Reactogenicity and Safety

In the 18–60 age cohort, participants were monitored for AEs until Day 42–45 after the 1st vaccination (Day 28–31 after the 2nd vaccination). The studied vaccine has shown good tolerability and safety. No deaths, serious AEs, or other significant AEs related to vaccination have been reported. All observed severe adverse events (SAEs) were classified as unlikely to be related or unrelated to the tested vaccine.

In total, during the study period, 244 AEs were recorded in 136 (34.2%) participants, of which 189 AEs were observed in 105 (35.2%) of the Vaccine Group participants and 55 AEs in 31 (31%) of the Placebo Group participants (Table 3, Appendix A).

The most common AE in the Vaccine Group was pain at the injection site. It was registered in the Placebo Group as well but was significantly less common (*p* < 0.05).

Changes in the parameters of clinical blood tests, biochemical blood tests and general urine analysis, observed at different time points, in most cases, were regarded as clinically insignificant and unrelated to the vaccination (Appendix A). Clinically significant deviations in the parameters of creatine phosphokinase, ALT, AST, CRP and leukocytes were observed in both groups, and their frequencies did not differ between the study groups. No clinically significant deviations in the parameters evaluated during physical examination, neurological status and ECG were observed in both groups.

One participant (0.3%, 0.1–1.9 95% CI) from the Vaccine Group died on Day 29 of the observation period as a result of an acute circulatory disorder. Autopsy examination revealed signs of chronic mild hepatitis, pre-existing stromal cardiosclerosis and dilated cardiomyopathy manifested by myocardial hypertrophy. The character of the lesions observed at histological examination indicated long-lasting pathology of the myocardium. As the participant had no clinically significant deviations in any of the clinical parameters throughout the observation period, including physical condition, body temperature, vital signs, urine analysis, biochemical parameters of blood, complete blood count, coagulation parameters and ECG, and had no local or systemic reactions after each vaccination, the relation of this SAE to vaccination was classified by the treating physician as unlikely.

In the 60+ age cohort, participants were monitored for AEs until Day 21 after the 3rd vaccination. Overall, in the 60+ age cohort, 827 AEs were observed in 184 (92%) volunteers throughout the observation period up to 21 days after 3rd vaccination. 317 AEs in 132 (66.3%) volunteers, 223 AEs in 110 (55.3%) volunteers and 287 AEs in 117 (58.5%) volunteers were observed over 21 days after the administration of the first, second and third doses of the vaccine, correspondingly (Table 4, Appendix A). Several adverse events could occur simultaneously in the same volunteer.

The studied vaccine has shown good tolerability and safety. The most frequent AEs reported by volunteers in 32.3% of cases were systemic reactions (mild asthenia, chills or fatigue) and pain at the vaccine injection site, which resolved within 3 days after vaccination in 85% of cases. Clinically significant abnormalities in the laboratory and instrumental analyses accounted for 38.5% of registered AE cases, nervous system disorders (insomnia, headache, dizziness, head discomfort, syncope, somnolence) for 8% cases, viral infections (COVID-19, other upper respiratory tract infections, herpes zoster, nasal and oral herpes) for 3.7% cases, respiratory disorders not associated with infections (oropharyngeal pain, nasal congestion, cough, dyspnea, pharyngeal edema, sore throat, rhinorrhea and sneezing) for 4.6% cases, musculoskeletal and connective disorders (arthralgia, pain in the extremities, back pain, myalgia, musculoskeletal discomfort) for 4.6% cases.

In total, out of 827 registered AEs, 81% were classified as mild, 15.2% moderate and 3.7% severe. The outcome of AEs was “recovery without consequences” in 86.5% of all cases, “recovery with consequences” in 0.2% of cases, “have not yet recovered” in 4% of cases and “consequences unknown” in 9.2% of cases.

No deaths or other AEs that were considered of special concern due to their clinical significance were reported. Of the 827 AEs in 184 (92%) volunteers, 22 (11%) volunteers had 22 serious AEs. COVID-19 accounted for 81.8% of serious AEs, and the remaining AEs were positive tests for SARS-CoV-2, hypertensive crisis, and upper limb fracture. 41% of the registered serious AEs were mild, 31.8% moderate and 27.2% severe. The observed serious AEs were considered by investigators as unrelated (91%) or unlikely to be related (9%) to vaccination (Appendix A). In 91% of cases, the outcome of serious AEs was “recovery without consequences”, in 4.5%, “recovery with consequences”, and in 4.5%, “consequences unknown”.

Changes in the parameters of clinical and biochemical blood tests and general urine analysis, registered in the dynamics of observation, were considered by the principal investigator in most cases as clinically insignificant and unrelated to vaccination, indicating the safety of the vaccine. Clinically significant changes in clinical and biochemical blood tests were noted in a few cases for each parameter, and in general, urine analysis only for erythrocyte and leukocyte levels.

Routine observations revealed no cases of clinically significant deviations in physical examination parameters among the study participants in the 60+ age cohort.

Most of the reported AEs where the relation to vaccination was considered “definite”, “probable” or “possible”, were the AEs typically expected after vaccination due to the effect of the vaccine. No vaccine-related AEs of particular concern, which were previously reported for other COVID-19 vaccine preparations (Guillain–Barré syndrome, generalized seizures, anaphylaxis, thrombocytopenia, coagulopathy, etc.) were reported in the volunteers that received CoviVac in the 60+ age cohort.

### 3.3. CoviVac Humoral Immunogenicity Assessment

#### 3.3.1. 18–60 Age Cohort

Immunogenicity assessment in the 18–60 age cohort was performed during Stage 3 on 122 participants of the Vaccine Group and in 45 participants of the Placebo Group. Due to the lack of defined correlates of protection against COVID-19 and the low correlation between seropositivity rates assessed in NT and CMIA at all time points (Appendix A, Appendix A), only NT results were used for calculations of seroconversion rates. In total, 28 days after the 2nd vaccination, 113/122 (92.6%) participants in the Vaccine Group and 18/45 (40.9%) participants in the Placebo Group were anti-SARS-CoV-2 nAB positive (Table 5).

Immunogenicity assessment was also performed separately both for the participants who were anti-SARS-CoV-2 nAB negative (69/122 in Vaccine Group and 28/45 in Placebo Group) or anti-SARS-CoV-2 nAB positive (53/122 in Vaccine Group and 17/45 in Placebo Group) at screening.

Participants with undetectable anti-SARS-CoV-2 antibodies at screening (with any method) showed increasing nAB titers and seroconversion from day 7 post 1st vaccination (Figure 3A). By Day 7 after the 2nd vaccination (21 days total) the seroconversion rate reached 84.1% and did not decrease during the period of observation (until Day 28 post 2nd vaccination, Figure 3B). GMT of nABs peaked at Day 21 after the 1st vaccination and did not decrease significantly during the period of observation (Figure 3A). Among the participants, who did not have detectable anti-SARS-CoV-2 nAB antibodies at screening, nABs were detected in 0/69 (0%), 15/69 (21.7%), 40/69 (57.9%), 58/69 (84.1%), 59/69 (85.5%) and 60/69 (86.9%) at screening, Days 7, 14, 21, 28 and 42, respectively. The GMT at Days 7, 14, 21, 28 and 42 were 1:2, 1:4, 1:10, 1:15 and 1:20, respectively (Figure 3A,B). When calculated for positive samples only GMTs at Days 7, 14, 21, 28 and 42 were 1:10, 1:13, 1:15, 1:23 and 1:30, respectively.

According to the protocol of the clinical trial, seroconversion for participants who had detectable SARS-CoV-2 nABs at screening was defined as 4-fold increase in nAB titers after vaccination compared to baseline titers.

Participants who had detectable antibodies against SARS-CoV-2 at screening showed increased nAB titers at days 7 and 14 post 1st vaccination. However, after the 2nd vaccination the titers did not increase significantly (Figure 3C). Moreover, vaccination of the participants with lower nAB titers (below 1:256) had a more pronounced effect of titer increase (4-fold and more), whereas we observed almost no booster effect in participants with high baseline nAB titers (1:512 and above) (Figure 3C). Nevertheless, at day 28 post 2nd vaccination in participants who had nAB titers below 1:256 at screening the rate of 4-fold increase in nAB levels was comparable to the seroconversion rates for participants, who were seronegative at screening (85.2% vs. 86.9%, correspondingly, Figure 3D).

#### 3.3.2. 60+ Age Cohort

Immunogenicity assessment in the 60+ age cohort was performed on all 199 volunteers who received at least one dose of the vaccine. At the time of inclusion in the study, the proportion of volunteers seropositive in the NT was 65.5%. The results of immunogenicity assessment in the 60+ group are shown in Table 6.

Despite decreased immune status in elderly individuals due to the natural aging process, CoviVac vaccination led to high immunological efficacy in individuals in the 60+ age cohort. Then, 21 days after triple vaccination, the seroconversion rate (defined as a fourfold nAB titer increase) was 72.6% in all volunteers, 88.5% in volunteers who were seronegative at screening, and 65.5% in volunteers who were seropositive at screening, with GMT of 145.76, 31.13, and 291.22, respectively.

In the same volunteers 21 days after the second dose of the vaccine, the seroconversion rate in seronegative individuals was 81.8%, while there was an insufficient immune response in volunteers who were seropositive at screening (29.2%) and in general in all volunteers regardless of baseline status (45.7%) with GMTs of 31.13, 291.22 and 86.6, respectively.

Therefore, the most preferable immunization regimen for persons over 60 years of age who are at risk for morbidity and mortality from COVID-19 is triple vaccination with 21-day intervals, which showed higher immunological efficacy.

### 3.4. Cellular Immunity Assessment

Parameters of cellular immunity were determined for the same 27 randomly picked participants from the Vaccine Group in Stage 3 of the phase II study in the 18–60 age cohort. Cellular immune response was assessed in the whole blood of the trial participants using QuantiFERON SARS-CoV-2 Starter Set (Qiagen, Germany) by stimulation of IFN-γ production by S-protein-specific CD4 and CD4/CD8 T cells using two SARS-CoV-2 S protein peptide pools (Ag1 and Ag2, respectively). The data are summarized in Figure 4.

Compared to baseline levels, we observed a statistically significant (*p* < 0.05) increase in IFN-γ production by peptide-stimulated CD4+ cells of vaccinated participants at Days 14 and 21 after the 1st vaccination and a statistically significant increase in combined IFN-γ production by stimulated CD4+ and CD8+ cells at Day 21 after the 1st vaccination, indicating the development of both SARS-CoV-2 S protein-specific CD4 helper and CD8 cytotoxic cells.

## 4. Discussion

According to the WHO, so far, 18 inactivated vaccines against COVID-19 have passed Phase I-II clinical trials worldwide, of which five have also passed Phase III trials (WHO, 2023). Successful clinical trials of 11 inactivated vaccines demonstrate that they are highly useful in the prevention of severe COVID-19 cases in the situation of a drastic global shortage of COVID-19 vaccines. Previously, the Chumakov Center has successfully developed two inactivated vaccines against tick-borne encephalitis and poliomyelitis produced using Vero cells [18,19]. Therefore, the development of the COVID-19 vaccine CoviVac was based on an established cell-cultivating platform using previously proven methods. Immunogenicity, safety and protective efficacy of CoviVac have been previously shown in preclinical studies [16].

Here, we present the results of clinical trials of the tolerability, safety, and immunogenicity of the inactivated whole virion concentrated purified coronavirus vaccine CoviVac in adult volunteers aged 18–60 (phases I/II) and older (phase IIb). The primary outcomes for safety assessment were the frequency and severity of AEs within the observation period, and the primary immunogenicity endpoints were the geometric mean nAB titer (GMT) and seroconversion rate (defined as fourfold or higher increase in GMT of specific antibodies in the NT) following complete vaccination regimen in each cohort.

### 4.1. CoviVac Safety in Phases I/II and IIb

Overall, in phases I/II starting from 3 October 2020, 498 volunteers were screened for eligibility at three clinical sites, and 400 volunteers aged 18–60 years were enrolled in the study. All participants were randomized into two groups: 300 participants were assigned to receive the vaccine and 100 participants were assigned to receive a placebo (two injections, 0.5 mL, 14 day intervals for both the vaccine and placebo).

The studied vaccine has shown good tolerability and safety. No deaths, serious adverse events, or other significant AEs related to vaccination have been reported. The most common AE in the Vaccine Group was pain at the injection site. It was registered in the Placebo Group as well but was significantly less common (*p* < 0.05). One participant (0.3%, 0.1–1.9 95% CI) from the Vaccine Group died on Day 29 of the observation period as a result of acute circulatory disorder, which was not related to the tested vaccine. Autopsy examination revealed signs of chronic mild hepatitis, pre-existing stromal cardiosclerosis and dilated cardiomyopathy manifested by myocardial hypertrophy, which were undetectable by the methods used at pre-vaccination screening.

The phase IIb study randomized 200 volunteers aged 60 years and older (mean age 67.26 ± 6.29 years). Of these, 199 volunteers received the CoviVac vaccine three times with an interval of 21 days. In total, 165 (82.5%) volunteers completed the study according to the Protocol.

In the 60+ age cohort, participants were monitored for AEs until Day 21 after the third vaccination. The studied vaccine has shown good tolerability and safety. No deaths, serious AEs or other AEs that were considered of special concern due to their clinical significance were reported. The most frequent AEs reported by volunteers were similar to the ones reported by the participants of the phase I/II study: 32.3% of cases were systemic reactions (mild asthenia, chills or fatigue) and pain at the vaccine injection site, which resolved within 3 days after vaccination in 85% of cases.

Therefore, overall, 497 clinical trial participants were vaccinated with CoviVac and included in the safety assessment. No serious adverse events or other significant AEs related to vaccination have been reported. The facts show a good safety profile of the CoviVac vaccine. The vaccine can be used for vaccination of adults over 18 years of age. We can anticipate the detection of some extremely rare AEs during the wide use of the vaccine, although the long-term history of use of other inactivated vaccines has proven their exceptional safety.

### 4.2. CoviVac Immunogenicity in Phases II (18–60 Years) and IIb (60+ Years)

The phase I/II study in healthy individuals was conducted from 3 October 2020 to 29 March 2021, during the spread of SARS-CoV-2 variant Alpha in the Russian Federation (with a maximum incidence of 29,935 cases per day) and implementation of restriction measures (isolation, use of PPE, etc.). Phase IIb was carried out in the high-risk group (participants over 60 years of age with chronic concomitant diseases) from 8 November 2021 to 17 June 2022, when variant Delta was substituted by variant Omicron, with the latter prevailing most of the time (maximum incidence of 203,949 people per day) against the background of a stable regress of the restrictive measures [20].

In the phase II study, immunogenicity assessment was performed by NT in 167 volunteers (122 in Vaccine Group and 45 in Placebo Group) separately for the participants who were anti-SARS-CoV-2 nAB negative (69/122 in the Vaccine Group and 28/45 in the Placebo Group) or anti-SARS-CoV-2 nAB positive (53/122 in the Vaccine Group and 17/45 in the Placebo Group) at screening. In the phase IIb study, immunogenicity assessment was performed by NT for all 199 volunteers who received at least one dose of the CoviVac vaccine, although 165 (82.5%) volunteers completed the study according to the Protocol.

Study participants from different age cohorts were vaccinated according to different schedules: phase II participants (aged 18–60) received two doses of the vaccine with 14 day intervals, whereas the phase IIb participants (aged 60+) received three doses with 21 day intervals. Changes in vaccination schedules were made to provide the most favorable conditions of anti-SARS-CoV-2 immunity development in 60+ recipients, who are expected to have declined immune response. The results of phase II showed an 81.8% seroconversion rate 21 days after the 2nd vaccination (in seronegative vaccinees). To achieve the same rates in the 60+ cohort, we increased the time between vaccinations and added a 3rd vaccination. This allowed us to achieve seroconversion in 88.5% of participants 21 days after the 3rd vaccination in phase IIb.

It is obvious that an increase in time between vaccinations and an additional dose of the vaccine would favor the phase II participants as well. However, CoviVac development and phase II were conducted in 2020 during the rise of the COVID-19 pandemic. The vaccine providing anti-SARS-CoV-2 immunity in a short period of time was extremely needed. Therefore, the shortest appropriate schedule was chosen for phase II and following use. Nevertheless, in the present time a three-vaccinations schedule with 21 days between injections seems appropriate.

Both for phase II and phase IIb, ‘seroconversion’ for the participants who were seropositive at screening was defined as a fourfold increase in nAB titers compared to baseline. The seroconversion rate for the participants of phase II who had NT titers in NT below 1:256 at screening was comparable to that for participants who were seronegative at screening (85.2% and 86.9%, respectively), while in the participants with nAB titers over 1:256 the seroconversion rate did not exceed 45%, for the participants who were seropositive at screening, the 2nd vaccination did not lead to a further significant increase in NT titers. Both facts signify that there is a maximum level of antibodies against SARS-CoV-2, which the human immune system can produce, and further antigen addition does not increase immunity. Several studies have shown that antibody responses generated both after vaccination and after SARS-CoV-2 infection can decrease significantly over time [21,22,23], which would require boosting vaccinations in the long term. However, subjects with pre-existing nAB in high titers cannot respond properly to a booster immunization, and in these cases, booster immunization can be inefficient. Therefore, testing of pre-existing immunity may be required before booster vaccination to understand its necessity.

Nevertheless, on Day 42 after the 1st vaccination, the seroconversion rate in participants of phase IIb who were seronegative at screening was 86.9% with an average nAB GMT of 1:20 (or 1:30 for positive samples only). However, these nAB levels are above the estimated levels providing protection from symptomatic SARS-CoV-2 infection [24]; further protectivity assessment must be performed in phase III trials.

In the phase IIb study, the percentage of participants seropositive at screening (65.5%, Table 6) was higher than in the phase I/II study (41.9%, Table 5), which is to be expected. Seroconversion rates in the 18–60 and 60+ cohorts were similar: in the phase II study, seroconversion was observed in 86.9% of seronegative participants, and in the phase IIb study, seroconversion was observed in 81.8% of seronegative participants after two vaccinations and in 88.5% after three vaccinations. In participants seropositive at screening, the seroconversion in phase II was observed in 35.8% cases, while in phase IIb, it was observed in 29.2% of cases after two vaccinations and 65.5% of cases after three vaccinations (Table 7). The fact supports the observation described above that if the titers of pre-existing anti-SARS-CoV-2 nABs in vaccinees are rather high (over 1:256), additional vaccination does not lead to a significant further increase in nAB levels.

Nevertheless, these data indicate pronounced dynamics in nAB development in phases II and IIb participants vaccinated with the vaccine CoviVac, in comparison with the phase II participants injected with a Placebo. Moreover, we did not observe a decline in humoral immunity development in phase IIb with individuals aged 60+: CoviVac vaccination was equally effective in terms of seroconversion (z = 5.180, *p* < 0.001) in participants aged 18–60 and 60+. A standard vaccination schedule of two injections of the vaccine CoviVac with an interval of 21 days led to the development of nABs in 96.4% of seronegative participants aged 60+ by the 42nd day after the first vaccination. An additional dose of the vaccine did not drastically improve the seroconversion rate among initially seronegative recipients (from 81.8 to 88.5%). However, we observed an increase in seroconversion rates among initially seropositive recipients from 29.2% 21 days post 2nd vaccination to 65.5% 21 days post 3rd vaccination. From these, we can assume that a standard schedule of two vaccinations can be used for immunization of 60+ recipients.

Moreover, during the phase II study, cellular immune response was assessed based on the production of IFN-γ by the CD4+ and CD8+ lymphocytes after stimulation with SARS-CoV-2 S-protein peptide pools. Compared to baseline levels, we observed a statistically significant (*p* < 0.05) increase in IFN-γ production by peptide-stimulated CD4+ cells in vaccinated participants at Days 14 and 21 after the 1st vaccination and a statistically significant increase in combined IFN-γ production by stimulated CD4+ and CD8+ cells at Day 21 after the 1st vaccination. These observations support the previously reported S protein-specific IFN-γ production by T-cells of individuals after two doses of inactivated whole virion adjuvanted COVID-19 vaccine [25].

### 4.3. Prospects for Further Development

Some serum samples collected during phase II were tested against strains of later circulating SARS-CoV-2 variants: Delta and Omicron (BA.2). Overall, 57.1% and 61.9% of samples neutralized Delta and Omicron strains [26], respectively. These results are in line with the outcomes of previous experiments confirming that CoviVac contains a whole virion with trimeric intact spikes [27], thus vaccination with CoviVac induces a wide range of antibodies, including cross-neutralizing epitopes of the S protein. We expect a further decline in the immunity against novel Omicron variants. However, boosting with vaccines based on the novel variants can be a solution, maintaining neutralizing antibodies developed as a response to a primary CoviVac immunization.

CoviVac production platform uses an infectious virus primarily isolated from the natural viral population multiplied in Vero cells expressing ACE2 receptor [16]. Therefore, this platform can be easily modified to use other SARS-CoV-2 variants as vaccine antigens. Such modified vaccines can be used as heterologous boosters in the ever-changing epidemiology of COVID-19.

## 5. Conclusions

In conclusion, inactivated vaccine CoviVac has shown good tolerability and safety, with over 85% NT seroconversion rates after complete vaccination course in participants who were seronegative at screening in both age groups: 18–60 and 60+. In participants, who were seropositive at screening and had nAB titers below 1:256, a single vaccination led to a fourfold increase in nAB levels in 85.2% of cases. These findings indicate that CoviVac can be successfully used both for primary vaccination in a two-dose regimen and for booster vaccination as a single dose in individuals with reduced neutralizing antibody levels.

## Figures and Tables

**Figure 1 viruses-15-01828-f001:**
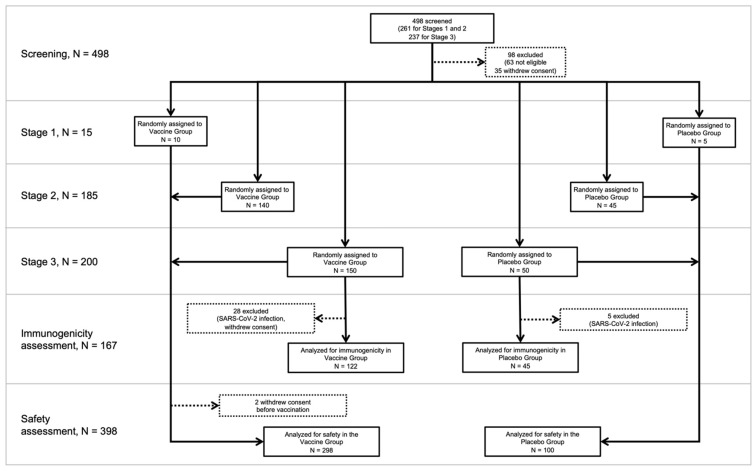
Study profile in 18–60 age cohort. Horizontal light grey lines represent the Stages of the study. Dotted lines represent exclusion of the participants from a given Stage of the study.

**Figure 2 viruses-15-01828-f002:**
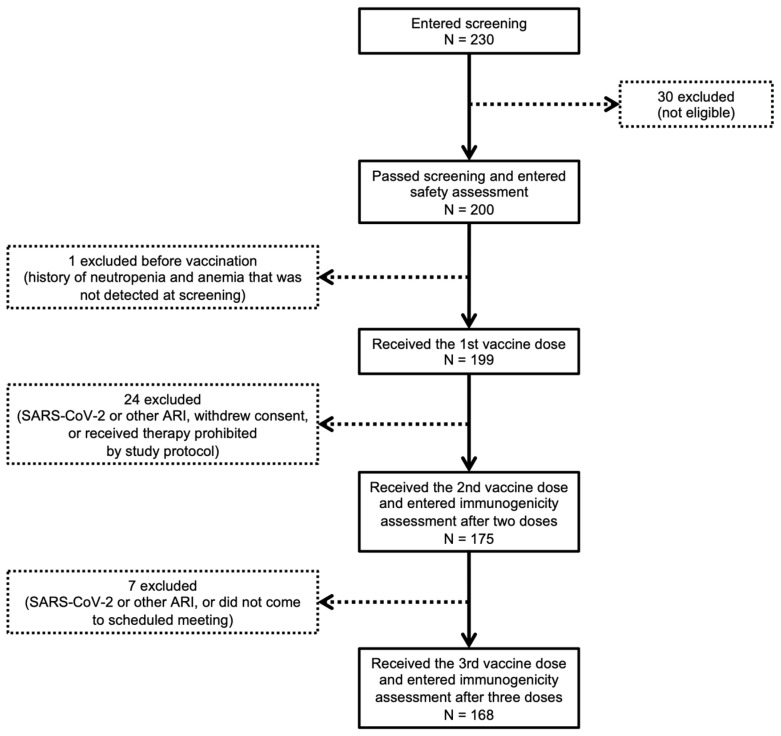
Study profile in 60+ age cohort. ARI—acute respiratory infection.

**Figure 3 viruses-15-01828-f003:**
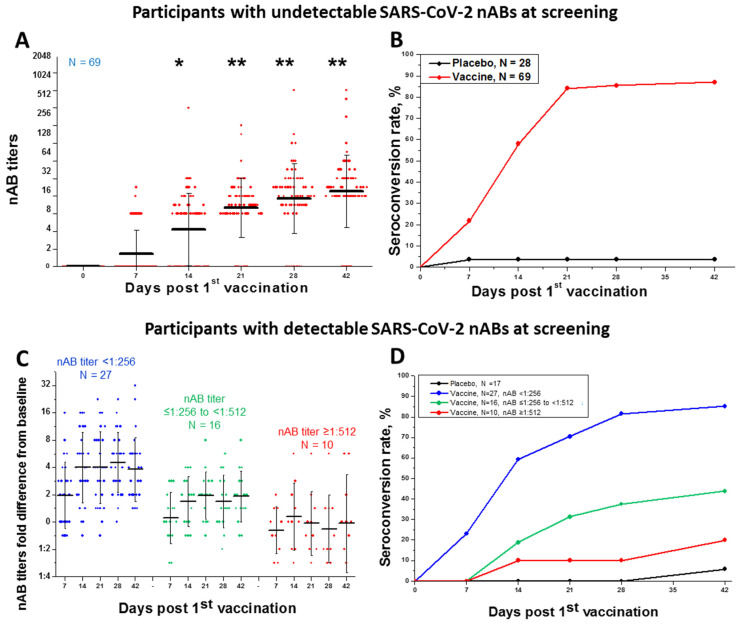
Seroconversion rates and SARS-CoV-2 neutralizing antibody (nAB) levels in sera of study participants in the 18–60 cohort at different time points post-vaccination. SARS-CoV-2 nAB titers (**A**) and cumulative seroconversion rate (**B**) in Vaccine Group and Placebo Group participants who were seronegative at screening. SARS-CoV-2 nAB titers fold difference from baseline level (**C**) and cumulative seroconversion rate (defined as 4-fold increase in the nAB levels) (**D**) in Vaccine Group and Placebo Group participants who had detectable nABs at screening at titers <1:256 (N = 27), ≤1:256–<1:512 (N = 16) or ≥1:512 (N = 10). Black line—Mean, whiskers—SD, *—titers at Days 14, 21, 28 and 42 significantly differ from baseline and from Day 7 [*p* < 0.05], **—the difference between titers at Days 14, 21, 28 and 42 is insignificant [*p* > 0.05].

**Figure 4 viruses-15-01828-f004:**
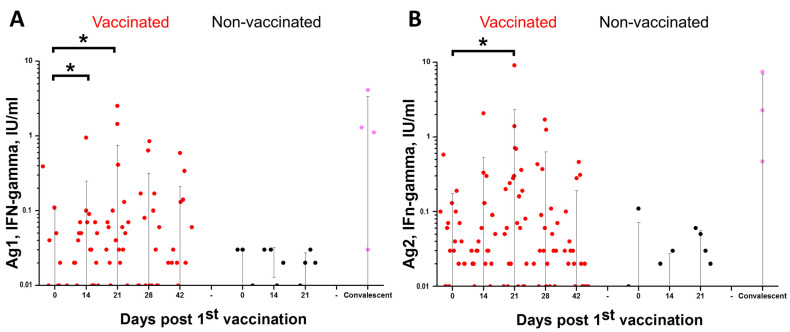
Anti-SARS-CoV-2 cellular immunity test results for randomly picked participants with undetectable SARS-CoV-2 nAB at screening at different time points post vaccination: (**A**) IFN-gamma levels induced by QuantiFERON Ag1; (**B**) IFN-gamma levels induced by QuantiFERON Ag2 (background was subtracted for each sample). Whiskers—SD; *—difference is significant (Mann–Whitney, *p* < 0.05).

**Table 1 viruses-15-01828-t001:** Demographic and anthropometric characteristics of the study participants in 18–60 age cohort.

Demographic and Anthropometric Parameters		Total Participants	Vaccine	Placebo
		n (%)		
**Safety assessment (Stages 1–3)**				
No. of participants		398	298	100
Age, years	Mean ± SD	33.2 ± 11.8	33.1 ± 11.8	33.5 ± 12.1
Sex	F	144 (36.2)	110 (36.9)	34 (34)
	M	254 (63.8)	188 (63.1)	66 (66)
BMI, kg/m^2^	Mean ± SD	23.9 ± 2.8	23.8 ± 2.8	24.0 ± 2.7
**Immunogenicity assessment (Stage 3)**				
No. of participants		167	122	45
Age, years	Mean ± SD	34.4 ± 12.9	34.6 ± 12.9	33.6 ± 13.3
Sex	F	61 (36.5)	46 (37.7)	15 (33.3)
	M	106 (63.5)	76 (63.3)	30 (66.7)
BMI, kg/m^2^	Mean ± SD	24.1 ± 2.7	24.1 ± 3.2	24.1 ± 2.7

**Table 2 viruses-15-01828-t002:** Demographic and anthropometric characteristics of the study participants in 60+ age cohort.

Demographic and Anthropometric Parameters		Total Participants
		n (%)
No. of participants		200
Age, years	Mean ± SD	67.26 ± 6.29
Sex	F	130 (65)
	M	70 (35)
BMI, kg/m^2^	Mean ± SD	27.09 ± 4.83

**Table 3 viruses-15-01828-t003:** Severity of adverse events in the Vaccine Group and in the Placebo Group within 28 days following each vaccination in 18–60 age cohort.

Adverse Event	Score	Vaccine Number of Participants with AEs (%, 95% CI)/Total Number of AEs			Placebo Number of Participants with AEs (%, 95% CI)/Total Number of AEs		
		Total Participants N = 298	Within 28 Days After		Total Participants N = 100	Within 28 Days After	
			First Dose	Second Dose		First Dose	Second Dose
**Local and systemic reactions**	Mild	57 (19.1%. 15.1–24.0%)/80	33 (11.1%. 8.0–15.1%)/37	34 (11.4%. 8.3–15.5%)/43	16 (16%. 10.1–24.4%)/23	7 (7%. 3.4–13.7%)/8	10 (10%. 5.5–17.4%)/15
	Moderate	3 (1%. 0.3–2.9%)/3	2 (0.7%. 0.2–2.4%)/2	1 (0.3%. 0.1–1.9%)/1			
	Severe	1 (0.3%. 0.1–1.9%)/1	-	1 (0.3%. 0.1–1.9%)/1			
Pain at the injection site	Mild	47 (15.8%. 12.1–20.3%)/55 *	30 (10.1%. 7.1–14.0%)/30	25 (8.4%. 5.7–12.1%)/25	11 (11%. 6.3–18.6%)/12 *	5 (5%. 2.2–11.2%)/5	7 (7%. 3.4–13.7%)/7
	Moderate	1 (0.3%. 0.1–1.9%)/1	1 (0.3%. 0.1–1.9%)/1				
Induration at the injection site	Mild	3 (1%. 0.3–2.9%)/3	1 (0.3%. 0.1–1.9%)/1	2 (0.7%. 0.2–2.4%)/2	3 (3%. 1.0–8.5%)/3	-	3 (3%. 1.0–8.5%)/3
Hematoma at the injection site	Mild	-	-	-	1 (1%. 0.2–5.4%)/1	1 (1%. 0.2–5.4%)/1	-
Swelling at the injection site	Mild				1 (1%. 0.2–5.4%)/1		1 (1%. 0.2–5.4%)/1
Itching at the injection site	Mild				1 (1%. 0.2–5.4%)/1		1 (1%. 0.2–5.4%)/1
Fever	Mild	11 (3.7%. 2.1–6.5%)/12	5 (1.7%. 0.7–3.9%)/5	6 (2%. 0.9–4.3%)/7	2 (2%. 0.6–7.0%)/2	2 (2%. 0.6–7.0%)/2	-
	Moderate	2 (0.7%. 0.2–2.4%)/2	1 (0.3%. 0.1–1.9%)/1	1 (0.3%. 0.1–1.9%)/1	-	-	-
Malaise	Mild	4 (1.3%. 0.5–3.4%)/10	1 (0.3%. 0.1–1.9%)/1	3 (1%. 0.3–2.9%)/9	2 (2%. 0.6–7.0%)/3	-	2 (2%. 0.6–7.0%)/3
	Severe	1 (0.3%. 0.1–1.9%)/1	-	1 (0.3%. 0.1–1.9%)/1	-	-	-
**Laboratory methods**	Mild	19 (6.4%. 4.1–9.7%)/22	7 (2.3%. 1.1–4.8%)/7	12 (4%)/15	7 (7%. 3.4–13.7%)/7	2 (2%. 0.6–7.0%)/2	5 (5%. 2.2–11.2%)/5
	Moderate	3 (1%. 0.3–2.9%)/3	1 (0.3%. 0.1–1.9%)/1	2 (0.7%. 0.2–2.4%)/2			
Increased leukocyte count	Moderate	1 (0.3%. 0.1–1.9%)/1	1 (0.3%. 0.1–1.9%)/1	-	-	-	-
Positive SARS-CoV-2 PCR test	Mild	17 (5.7%. 3.6–8.9%)/17	7 (2.3%. 1.1–4.8%)/7	10 (3.4%. 1.8–6.1%)/10	6 (6%. 2.8–12.5%)/6	2 (2%. 0.6–7.0%)/2	4 (4%. 1.6–9.8%)/4
	Moderate	2 (0.7%. 0.2–2.4%)/2		2 (0.7%. 0.2–2.4%)/2			
Increased creatine phosphokinase level	Mild	1 (0.3%. 0.1–1.9%)/1	-	1 (0.3%. 0.1–1.9%)/1			
	Moderate	2 (0.7%. 0.2–2.4%)/2		2 (0.7%. 0.2–2.4%)/2	1 (1%. 0.2–5.4%)/1	-	1 (1%. 0.2–5.4%)/1
Increased ALT level	Mild	1 (0.3%. 0.1–1.9%)/1	-	1 (0.3%. 0.1–1.9%)/1	-	-	-
Increased AST level	Mild	1 (0.3%. 0.1–1.9%)/1	-	1 (0.3%. 0.1–1.9%)/1	-	-	-
Increased CRP level	Mild	1 (0.3%. 0.1–1.9%)/1	-	1 (0.3%. 0.1–1.9%)/1	-	-	-
**Blood and lymphatic system disorders**	Mild	1 (0.3%. 0.1–1.9%)/1	1 (0.3%. 0.1–1.9%)/1	-			
Inguinal lymphadenitis	Mild	1 (0.3%. 0.1–1.9%)/1	1 (0.3%. 0.1–1.9%)/1	-	-	-	-
**Blood vessel disorders**					1 (1%. 0.2–5.4%)/1	1 (1%. 0.2–5.4%)/1	-
Deep venous leg thrombosis	Moderate	-	-	-	1 (1%. 0.2–5.4%)/1	1 (1%. 0.2–5.4%)/1	-
**Infections and parasitic invasions**	Mild	22 (7.4%. 4.9–10.9%)/22	15 (5%. 2.2–11.2%)/15	7 (2.3%. 1.1–4.8%)/7	3 (3%. 1.0–8.5%)/3	2 (2%. 0.6–7.0%)/2	1 (1%. 0.2–5.4%)/1
	Moderate	7 (2.3%. 1.1–4.8%)/7	3 (1%. 0.3–2.9%)/3	4 (1.3%. 0.5–3.4%)/4			
	Severe				1 (1%. 0.2–5.4%)/1	-	1 (1%. 0.2–5.4%)/1
SARS-CoV-2 infection	Mild	20 (6.7%. 4.4–10.1%)/20	14 (4.7%. 2.8–7.7%)/14	6 (2%. 0.9–4.3%)/6	3 (3%. 1.0–8.5%)/3	2 (2%. 0.6–7.0%)/2	1 (1%. 0.2–5.4%)/1
	Moderate	5 (1.7%. 0.7–3.9%)/5	2 (0.7%. 0.2–2.4%)/2	3 (1%. 0.3–2.9%)/3	-	-	-
	Severe	-	-	-	1 (1%. 0.2–5.4%)/1	-	1 (1%. 0.2–5.4%)/1
Candidiasis	Mild	1 (0.3%. 0.1–1.9%)/1	-	1 (0.3%. 0.1–1.9%)/1	-	-	-
Other upper respiratory tract infections	Mild	1 (0.3%. 0.1–1.9%)/1	1 (0.3%. 0.1–1.9%)/1	-	-	-	-
	Moderate	2 (0.7%. 0.2–2.4%)/2	1 (0.3%. 0.1–1.9%)/1	1 (0.3%. 0.1–1.9%)/1	-	-	-
**Nervous system disorders**	Mild	11 (3.7%. 2.1–6.5%)/20	4 (1.3%. 0.5–3.4%)/4	9 (3%. 1.6–5.6%)/16	6 (6%. 2.8–12.5%)/12	3 (3%. 1.0–8.5%)/3	3 (3%. 1.0–8.5%)/9
Headache	Mild	11 (3.7%. 2.1–6.5%)/19	4 (1.3%. 0.5–3.4%)/4	8 (2.7%. 1.4–5.2%)/15	6 (6%. 2.8–12.5%)/12	3 (3%. 1.0–8.5%)/3	3 (3%. 1.0–8.5%)/9
Dizziness	Mild	1 (0.3%. 0.1–1.9%)/1	-	1 (0.3%. 0.1–1.9%)/1	-	-	-
**Musculoskeletal and connective tissue disorders**	Mild	4 (1.3%. 0.5–3.4%)/6	2 (0.7%. 0.2–2.4%)/3	2 (0.7%. 0.2–2.4%)/3			
	Moderate	1 (0.3%. 0.1–1.9%)/1		1 (0.3%. 0.1–1.9%)/2	1 (1%. 0.2–5.4%)/1	1 (1%. 0.2–5.4%)/1	-
Arthralgia	Mild	2 (0.7%. 0.2–2.4%)/2	1 (0.3%. 0.1–1.9%)/1	1 (0.3%. 0.1–1.9%)/1	-	-	-
	Moderate	1 (0.3%. 0.1–1.9%)/1	-	1 (0.3%. 0.1–1.9%)/1	-	-	-
Lumbar pain	Moderate	-	-	-	1 (1%. 0.2–5.4%)/1	1 (1%. 0.2–5.4%)/1	-
Myalgia	Mild	4 (1.3%. 0.5–3.4%)/4	2 (0.7%. 0.2–2.4%)/2	2 (0.7%. 0.2–2.4%)/2	-	-	-
	Moderate	1 (0.3%. 0.1–1.9%)/1	-	1 (0.3%. 0.1–1.9%)/1	-	-	-
**Gastrointestinal tract disorders**	Mild	3 (1%. 0.3–2.9%)/3	1 (0.3%. 0.1–1.9%)/1	2 (0.7%. 0.2–2.4%)/2	1 (1%. 0.2–5.4%)/1	-	1 (1%. 0.2–5.4%)/1
	Moderate	1 (0.3%. 0.1–1.9%)/1	-	1 (0.3%. 0.1–1.9%)/1			
Diarrhea	Mild	1 (0.3%. 0.1–1.9%)/1	-	1 (0.3%. 0.1–1.9%)/1	1 (1%. 0.2–5.4%)/1	-	1 (1%. 0.2–5.4%)/1
Pyrosis	Mild	1 (0.3%. 0.1–1.9%)/1	1 (0.3%. 0.1–1.9%)/1	-	-	-	-
Nausea	Mild	1 (0.3%. 0.1–1.9%)/1	-	1 (0.3%. 0.1–1.9%)/1	-	-	-
	Moderate	1 (0.3%. 0.1–1.9%)/1	-	1 (0.3%. 0.1–1.9%)/1	-	-	-
**Hearing disorders**	Mild	1 (0.3%. 0.1–1.9%)/1	1 (0.3%. 0.1–1.9%)/1	-			
Ear congestion	Mild	1 (0.3%. 0.1–1.9%)/1	1 (0.3%. 0.1–1.9%)/1	-	-	-	-
**Disorders of respiratory system and mediastinal organs**	Mild	6 (2%. 0.9–4.3%)/6	2 (0.7%. 0.2–2.4%)/2	4 (1.3%. 0.5–3.4%)/4	3 (3%. 1.0–8.5%)/5	1 (1%. 0.2–5.4%)/1	2 (2%. 0.6–7.0%)/4
	Moderate	1 (0.3%. 0.1–1.9%)/1		1 (0.3%. 0.1–1.9%)/1			
Throat pain	Mild	1 (0.3%. 0.1–1.9%)/1	-	1 (0.3%. 0.1–1.9%)/1	1 (1%. 0.2–5.4%)/2	-	1 (1%. 0.2–5.4%)/2
	Moderate	1 (0.3%. 0.1–1.9%)/1		1 (0.3%. 0.1–1.9%)/1			
Pain in the oropharynx	Mild	1 (0.3%. 0.1–1.9%)/1	-	1 (0.3%. 0.1–1.9%)/1	1 (1%. 0.2–5.4%)/1	-	1 (1%. 0.2–5.4%)/1
Cough	Mild	1 (0.3%. 0.1–1.9%)/1	-	1 (0.3%. 0.1–1.9%)/1	1 (1%. 0.2–5.4%)/1	-	1 (1%. 0.2–5.4%)/1
Impaired sense of smell	Mild	1 (0.3%. 0.1–1.9%)/1	1 (0.3%. 0.1–1.9%)/1	-	-	-	-
Dyspnea	Mild	1 (0.3%. 0.1–1.9%)/1	-	1 (0.3%. 0.1–1.9%)/1	-	-	-
Sore throat	Mild	1 (0.3%. 0.1–1.9%)/1	1 (0.3%. 0.1–1.9%)/1	-	1 (1%. 0.2–5.4%)/1	1 (1%. 0.2–5.4%)/1	-
**Metabolic disorders**	Mild	2 (0.7%. 0.2–2.4%)/8	-	2 (0.7%. 0.2–2.4%)/8	1 (1%. 0.2–5.4%)/1	-	1 (1%. 0.2–5.4%)/1
	Moderate	1 (0.3%. 0.1–1.9%)/1	-	1 (0.3%. 0.1–1.9%)/1			
Impaired appetite	Mild	2 (0.7%. 0.2–2.4%) /8	-	2 (0.7%. 0.2–2.4%)/8	1 (1%. 0.2–5.4%)/1	-	1 (1%. 0.2–5.4%)/1
	Moderate	1 (0.3%. 0.1–1.9%)/1	-	1 (0.3%. 0.1–1.9%)/1	-	-	-
**Other**	Severe	1 (0.3%. 0.1–1.9%)/1	-	1 (0.3%. 0.1–1.9%)/1			
Death from acute circulatory disorder **	Severe	1 (0.3%. 0.1–1.9%)/1	-	1 (0.3%. 0.1–1,9%)/1			

* Differences are statistically significant (*p* < 0.05); ** Relation to vaccination classified as unlikely by the treating physician; - AE was not detected.

**Table 4 viruses-15-01828-t004:** Severity of adverse events within 21 days following each vaccination in the 60+ age cohort.

Adverse Event	Score	Number of Participants with AEs (%)/Number of AEs			
		Within 21 Days After			Total Participants, N = 199
		First Dose	Second Dose	Third Dose	
**Local and systemic reactions**	Mild	56 (28%)/85	45 (22.5%)/74	32 (16%)/55	79 (39.5%)/214
	Moderate	19 (9.5%)/26	8 (4%)/8	11 (5.5%)/13	23 (11.5%)/47
	Severe	3 (1.5%)/4	2 (1%)/2	–	5 (2.5%)/6
Asthenia	Mild	4 (2%)/4	1 (0.5%)/1	–	5 (2.5%)/5
Pain	Mild	–	1 (0.5%)/1	2 (1%)/2	3 (1.5%)/3
Pain at the injection site	Mild	43 (21.5%)/43	35 (17.5%)/36	17 (8.5%)/17	62 (31%)/96
	Moderate	10 (5%)/10	5 (2.5%)/5	7 (3.5%)/7	16 (8%)/22
	Severe	3 (1.5%)/3	2 (1%)/2	–	5 (2.5%)/5
Itching at the injection site	Mild	4 (2%)/4	3 (1.5%)/3	7 (3.5%)/7	11 (5.5%)/14
	Moderate	–	1 (0.5%)/1	–	1 (0.5%)/1
Chills	Mild	4 (2%)/5	3 (1.5%)/3	1 (0.5%)/1	7 (3.5%)/9
	Moderate	2 (1%)/2	–	–	2 (1%)/2
Swelling at the injection site	Mild	1 (0.5%)/1	2 (1%)/3	3 (1.5%)/3	5 (2.5%)/7
Foreign body sensation	Mild	1 (0.5%)/1	–	–	1 (0.5%)/1
Fever	Mild	3 (1.5%)/3	1 (0.5%)/1	3 (1.5%)/3	7 (3.5%)/7
	Moderate	1 (0.5%)/1	–	–	1 (0.5%)/1
Malaise	Mild	–	1 (0.5%)/1	–	1 (0.5%)/1
Induration at the injection site	Mild	3 (1.5%)/3	7 (3.5%)/8	6 (3%)/6	11 (5.5%)/17
	Moderate	1 (0.5%)/1	–	–	1 (0.5%)/1
Fatigue	Mild	15 (7.5%)/17	12 (6%)/15	10 (5%)/11	27 (13.5%)/43
	Moderate	10 (5%)/12	2 (1%)/2	5 (2.5%)/6	15 (12.5%)/20
	Severe	1 (0.5%)/1	–	–	1 (0.5%)/1
Discomfort at the injection site	Mild	1 (0.5%)/1	–	–	1 (0.5%)/1
Erythema at the injection site	Mild	3 (1.5%)/3	2 (1%)/2	5 (2.5%)/5	9 (4.5%)/10
**Laboratory methods**	Mild	68 (34%)/88	45 (22.5%)/57	72 (36%)/109	133 (66.5%)/254
	Moderate	9 (4.5%)/11	12 (6%)/13	19 (9.5%)/22	36 (18%)/46
	Severe	–	7 (3.5%)/7	9 (4.5%)/11	15 (12.5%)/18
Proteinuria	Mild	2 (1%)/2	–	3 (1.5%)/3	5 (2.5%)/5
	Moderate	1 (0.5%)/1	–	–	1 (0.5%)/1
Glycosuria	Severe	–	–	1 (0.5%)/1	1 (0.5%)/1
Leukocytes in urine	Mild	2 (1%)/2	3 (1.5%)/3	5 (2.5%)/5	9 (4.5%)/10
Abnormal ALT level	Moderate	3 (1.5%)/3	–	–	3 (1.5%)/3
Abnormal glucose level	Severe	–	–	1 (0.5%)/1	1 (0.5%)/1
Abnormal urea level	Moderate	–	1 (0.5%)/1	1 (0.5%)/1	2 (1%)/2
	Severe	–	–	1 (0.5%)/1	1 (0.5%)/1
Abnormal CRP level	Mild	1 (0.5%)/1	1 (0.5%)/1	1 (0.5%)/2	3 (1.5%)/4
	Moderate	–	–	3 (1.5%)/3	3 (1.5%)/3
Abnormal cholesterol level	Mild	–	–	1 (0.5%)/1	1 (0.5%)/1
	Moderate	–	–	2 (1%)/2	2 (1%)/2
Abnormal blood ALP level	Mild	–	–	2 (1%)/3	2 (1%)/3
	Moderate	–	3 (1.5%)/3	–	3 (1.5%)/3
Increased arterial blood pressure	Mild	4 (2%)/5	2 (1%)/2	1 (0.5%)/2	4 (2%)/9
	Severe	–	1 (0.5%)/1	1 (0.5%)/1	2 (1%)/2
Increased diastolic blood pressure	Mild	5 (2.5%)/5	4 (2%)/4	1 (0.5%)/1	10 (5%)/10
Increased relative density of urine	Mild	–	–	1 (0.5%)/1	1 (0.5%)/1
Increased systolic blood pressure	Mild	5 (2.5%)/5	3 (1.5%)/3	1 (0.5%)/1	9 (4.5%)/9
Increased erythrocyte sedimentation rate	Mild	2 (1%)/2	1 (0.5%)/1	1 (0.5%)/1	4 (2%)/4
Increased ALT level	Mild	4 (2%)/4	1 (0.5%)/1	6 (3%)/7	11 (5.5%)/12
Increased AST level	Mild	5 (2.5%)/5	2 (1%)/2	6 (3%)/8	12 (6%)/15
	Moderate	–	–	1 (0.5%)/1	1 (0.5%)/1
Increased bilirubin level	Mild	2 (1%)/2	2 (1%)/2	3 (1.5%)/3	6 (3%)/7
	Moderate	1 (0.5%)/1	2 (1%)/2	2 (1%)/2	5 (2.5%)/5
Increased glucose level	Mild	2 (1%)/2	2 (1%)/2	3 (1.5%)/3	7 (3.5%)/7
	Moderate	2 (1%)/2	1 (0.5%)/1	4 (2%)/4	6 (3%)/7
	Severe	–	1 (0.5%)/1	–	1 (0.5%)/1
Increased creatinine level	Mild	1 (0.5%)/1	1 (0.5%)/1	1 (0.5%)/1	3 (1.5%)/3
Increased creatine phosphokinase level	Mild	3 (1.5%)/3	5 (2.5%)/5	7 (3.5%)/7	15 (7.5%)/16
	Moderate	2 (1%)/2	–	1 (0.5%)/1	3 (1.5%)/3
	Severe	–	1 (0.5%)/1	–	1 (0.5%)/1
Increased urea phosphokinase level	Mild	2 (1%)/2	3 (1.5%)/3	7 (3.5%)/7	11 (5.5%)/12
	Moderate	–	–	1 (0.5%)/1	1 (0.5%)/1
Increased CRP level	Mild	1 (0.5%)/1	2 (1%)/2	1 (0.5%)/1	4 (2%)/4
	Moderate	1 (0.5%)/1	1 (0.5%)/1	–	2 (1%)/2
Increased transaminase levels	Mild	–	1 (0.5%)/1	–	1 (0.5%)/1
Increased cholesterol level	Mild	4 (2%)/4	4 (2%)/4	10 (5%)/10	17 (8.5%)/18
	Moderate	–	4 (2%)/4	5 (2.5%)/5	9 (4.5%)/9
	Severe	–	4 (2%)/4	5 (2.5%)/5	9 (4.5%)/9
Increased ALP level	Mild	2 (1%)/2	1 (0.5%)/1	3 (1.5%)/3	5 (2.5%)/6
Increased basophil count	Mild	1 (0.5%)/1	–	1 (0.5%)/1	1 (0.5%)/2
Increased leukocyte count	Mild	6 (3%)/6	1 (0.5%)/1	–	7 (3.5%)/7
	Moderate	1 (0.5%)/1	–	–	1 (0.5%)/1
Increased eosinophil count	Mild	1 (0.5%)/1	2 (1%)/2	1 (0.5%)/1	4 (2%)/4
SARS-CoV-2 infection	Moderate	–	1 (0.5%)/1	–	1 (0.5%)/1
	Severe	–	–	1 (0.5%)/1	1 (0.5%)/1
Decreased hemoglobin level	Mild	–	–	4 (2%)/4	4 (2%)/4
Decreased glucose level	Mild	–	–	3 (1.5%)/3	3 (1.5%)/3
	Moderate	–	–	1 (0.5%)/1	1 (0.5%)/1
	Severe	–	–	1 (0.5%)/1	1 (0.5%)/1
Decreased creatine phosphokinase level	Mild	–	–	1 (0.5%)/1	1 (0.5%)/1
Decreased total protein level	Mild	–	–	2 (1%)/2	2 (1%)/2
Decreased lymphocyte count	Mild	2 (1%)/2	1 (0.5%)/1	–	3 (1.5%)/3
Decreased neutrophil count	Mild	8 (4%)/8	1 (0.5%)/1	2 (1%)/2	10 (5%)/11
Decreased platelet count	Mild	1 (0.5%)/1	1 (0.5%)/1	–	2 (1%)/2
Increased lymphocyte count	Mild	2 (1%)/2	–	–	2 (1%)/2
Cylindruria	Mild	–	1 (0.5%)/1	–	1 (0.5%)/1
Erythrocytes in urine	Mild	19 (9.5%)/19	12 (6%)/12	25 (12.5%)/25	52 (26%)/56
	Moderate	–	–	1 (0.5%)/1	1 (0.5%)/1
**Gastrointestinal tract disorders**	Mild	11 (5.5%)/11	3 (1.5%)/3	2 (1%)/3	14 (7%)/17
	Moderate	2 (1%)/2	–	1 (0.5%)/1	3 (1.5%)/3
Abdominal discomfort	Mild	1 (0.5%)/1	–	–	1 (0.5%)/1
Abdominal pain	Mild	1 (0.5%)/1	–	–	1 (0.5%)/1
Lip pain	Mild	1 (0.5%)/1	–	–	1 (0.5%)/1
Diarrhea	Mild	3 (1.5%)/3	2 (1%)/2	2 (1%)/2	6 (3%)/7
	Moderate	1 (0.5%)/1	–	1 (0.5%)/1	2 (1%)/2
Dyspepsia	Mild	2 (1%)/2	–	–	2 (1%)/2
Toothache	Mild	–	1 (0.5%)/1	–	1 (0.5%)/1
Nausea	Mild	3 (1.5%)/3	–	1 (0.5%)/1	4 (2%)/4
	Moderate	1 (0.5%)/1	–	–	1 (0.5%)/1
**Infections and parasitic invasions**	Mild	5 (2.5%)/5	6 (3%)/6	6 (3%)/6	17 (8.5%)/17
	Moderate	3 (1.5%)/3	2 (1%)/2	4 (2%)/4	9 (4.5%)/9
	Severe	1 (0.5%)/1	1 (0.5%)/1	3 (1.5%)/3	5 (2.5%)/5
Upper respiratory tract infections	Mild	1 (0.5%)/1	4 (2%)/4	2 (1%)/2	10 (5%)/10
	Moderate	–	2 (1%)/2	1 (0.5%)/1	3 (1.5%)/3
SARS-CoV-2 infection	Mild	2 (1%)/2	2 (1%)/2	4 (2%)/4	8 (4%)/8
	Moderate	3 (1.5%)/3	–	2 (1%)/2	5 (2.5%)/5
	Severe	1 (0.5%)/1	1 (0.5%)/1	3 (1.5%)/3	5 (2.5%)/5
Nasal herpes simplex lesions	Mild	1 (0.5%)/1	–	–	1 (0.5%)/1
Herpes zoster	Moderate	–	–	1 (0.5%)/1	1 (0.5%)/1
Labial herpes simplex lesions	Mild	1 (0.5%)/1	–	–	1 (0.5%)/1
**Metabolic and nutrition disorders**	Mild	6 (3%)/6	1 (0.5%)/1	9 (4.5%)/10	14 (7%)/17
	Moderate	–	1 (0.5%)/1	2 (1%)/2	3 (1.5%)/3
Hyperglycemia	Mild	1 (0.5%)/1	1 (0.5%)/1	–	2 (1%)/2
Hypoglycemia	Mild	1 (0.5%)/1	–	7 (3.5%)/7	8 (4%)/8
	Moderate	–	1 (0.5%)/1	2 (1%)/2	3 (1.5%)/3
Hypoproteinemia	Mild	4 (2%)/4	–	3 (1.5%)/3	5 (2.5%)/7
**Disorders of respiratory system and mediastinal organs**	Mild	7 (3.5%)/7	9 (4.5%)/13	8 (4%)/10	21 (10.5%)/31
	Moderate	1 (0.5%)/1	–	1 (0.5%)/1	2 (1%)/2
Pain in the oropharynx	Mild	1 (0.5%)/1	2 (1%)/2	1 (0.5%)/1	4 (2%)/4
	Moderate	–	–	1 (0.5%)/1	1 (0.5%)/1
Nasal congestion	Mild	1 (0.5%)/1	2 (1%)/2	1 (0.5%)/1	4 (2%)/4
Cough	Mild	3 (1.5%)/3	1 (0.5%)/1	2 (1%)/2	6 (3%)/6
Dyspnea	Mild	–	1 (0.5%)/1	–	1 (0.5%)/1
Swelling of the pharynx	Mild	–	–	2 (1%)/2	2 (1%)/2
Sore throat	Mild	2 (1%)/2	1 (0.5%)/1	1 (0.5%)/1	4 (2%)/4
Rhinorrhea	Mild	1 (0.5%)/1	5 (2.5%)/5	3 (1.5%)/3	8 (4%)/9
	Moderate	1 (0.5%)/1	–	–	1 (0.5%)/1
Sneezing	Mild	–	1 (0.5%)/1	–	1 (0.5%)/1
**Skin and subcutaneous tissue disorders**	Mild	3 (1.5%)/3	1 (0.5%)/1	–	4 (2%)/4
Hyperhidrosis	Mild	1 (0.5%)/1	–	–	1 (0.5%)/1
Itching	Mild	2 (1%)/2	–	–	2 (1%)/2
Rash	Mild	–	1 (0.5%)/1	–	1 (0.5%)/1
**Muscular. skeletal and connective tissue disorders**	Mild	14 (7%)/15	8 (4%)/9	7 (3.5%)/8	23 (11.5%)/32
	Moderate	4 (2%)/4	–	2 (1%)/2	6 (3%)/6
Arthralgia	Mild	2 (1%)/2	2 (1%)/2	1 (0.5%)/2	5 (2.5%)/6
Pain in the limb	Mild	1 (0.5%)/1	–	1 (0.5%)/1	2 (1%)/2
Back pain	Mild	1 (0.5%)/1	–	–	1 (0.5%)/1
	Moderate	–	–	1 (0.5%)/1	1 (0.5%)/1
Myalgia	Mild	10 (5%)/10	7 (3.5%)/7	5 (2.5%)/5	19 (9.5%)/22
	Moderate	4 (2%)/4	–	1 (0.5%)/1	5 (2.5%)/5
Musculoskeletal discomfort	Mild	1 (0.5%)/1	–	–	1 (0.5%)/1
**Nervous system disorders**	Mild	20 (10%)/26	18 (9%)/22	9 (4.5%)/11	37 (18.5%)/59
	Moderate	1 (0.5%)/1	1 (0.5%)/1	3 (1.5%)/4	5 (2.5%)/6
	Severe	–	–	1 (0.5%)/1	1 (0.5%)/1
Insomnia	Mild	1 (0.5%)/1	2 (1%)/2	–	3 (1.5%)/3
Headache	Mild	15 (7.5%)/20	15 (7.5%)/18	8 (4%)/9	32 (16%)/47
	Moderate	1 (0.5%)/1	1 (0.5%)/1	3 (1.5%)/4	1 (0.5%)/1
	Severe	–	–	1 (0.5%)/1	1 (0.5%)/1
Dizziness	Mild	2 (1%)/2	1 (0.5%)/1	1 (0.5%)/1	4 (2%)/4
Discomfort in the head	Mild	–	1 (0.5%)/1	–	1 (0.5%)/1
Syncope	Mild	–	–	1 (0.5%)/1	1 (0.5%)/1
Somnolence	Mild	2 (1%)/3	–	–	2 (1%)/3
**Eye disorders**	Mild	1 (0.5%)/1	–	1 (0.5%)/1	1 (0.5%)/2
Hypersecretory lacrimation	Mild	1 (0.5%)/1	–	–	1 (0.5%)/1
Photopsy	Mild	–	–	1 (0.5%)/1	1 (0.5%)/1
**Hearing disorders**	Mild	1 (0.5%)/1	–	1 (0.5%)/1	2 (1%)/2
Ear congestion	Mild	1 (0.5%)/1	–	–	1 (0.5%)/1
tinnitus	Mild	–	–	1 (0.5%)/1	1 (0.5%)/1
**Renal and urinary tract disorders**	Mild	6 (3%)/6	–	3 (1.5%)/3	9 (5.5%)/9
	Moderate	–	1 (0.5%)/1	–	1 (0.5%)/1
	Severe	1 (0.5%)/1	–	–	1 (0.5%)/1
Hematuria	Mild	1 (0.5%)/1	–	–	1 (0.5%)/1
	Moderate	–	1 (0.5%)/1	–	1 (0.5%)/1
Glycosuria	Severe	1 (0.5%)/1	–	–	1 (0.5%)/1
Leukocyturia	Mild	4 (2%)/4	–	–	4 (2%)/4
Proteinuria	Mild	1 (0.5%)/1	–	3 (1.5%)/3	4 (2%)/4
**Heart disorders**	Mild	5 (2.5%)/5	–	3 (1.5%)/4	8 (4%)/9
	Moderate	1 (0.5%)/1	–	–	1 (0.5%)/1
Arrhythmia	Mild	–	–	1 (0.5%)/1	1 (0.5%)/2
	Moderate	1 (0.5%)/1	–	–	1 (0.5%)/1
Chest pain	Mild	1 (0.5%)/1	–	–	1 (0.5%)/1
Bradycardia	Mild	2 (1%)/2	–	–	2 (1%)/2
Discomfort in the heart	Mild	–	–	1 (0.5%)/2	1 (0.5%)/2
Stenocardia	Mild	1 (0.5%)/1	–	1 (0.5%)/1	2 (1%)/2
Tachycardia	Mild	1 (0.5%)/1	–	–	1 (0.5%)/1
**Vascular disorders**	Mild	1 (0.5%)/1	–	2 (1%)/2	3 (1.5%)/3
Hypertensive crisis	Mild	1 (0.5%)/1	–	–	1 (0.5%)/1
Blood vessel rupture	Mild	–	–	1 (0.5%)/1	1 (0.5%)/1
Nosebleed	Mild	–	–	1 (0.5%)/1	1 (0.5%)/1
**Injuries. intoxications and complications of procedures**	Moderate	1 (0.5%)/1	1 (0.5%)/1	–	2 (1%)/2
Fracture of the upper limb	Moderate	1 (0.5%)/1	–	–	1 (0.5%)/1
Meniscus injury	Moderate	–	1 (0.5%)/1	–	1 (0.5%)/1

**Table 5 viruses-15-01828-t005:** Seroconversion rates (by NT) in study participants in the 18–60 age cohort at different time points post vaccination.

Group	N	Positive, n (%, 95% CI)					
		Screening	Day 7	Day 14	Day 21 (7 Days after the 2nd Vaccination)	Day 28 (14 Days after the 2nd Vaccination)	Day 42 (28 Days after the 2nd Vaccination)
Vaccine	122	53 (43.4, 35.0–52.3)	68 (55.7, 46.9–64.2)	92 (75.4, 67.1–82.2)	112 (91.8, 85.6–95.5)	112 (91.8, 85.6–95.5)	113 (92.6, 86.6–96.1)
Placebo	45	17 (37.8, 25.1–52.4)	18 (40.0, 27.0–54.6)	19 (42.2, 29.0–56.7)	17 (38.6, 25.7–53.4) ^1^	17 (37.8, 25.1–52.4)	18 (40.9, 27.7–55.6) ^1^

^1^ N = 44.

**Table 6 viruses-15-01828-t006:** Seroconversion rates and SARS-CoV-2 neutralizing antibody (nAB) levels in sera of phase IIb study participants after vaccination.

Immunogenicity Parameters	Seroconversion Rate		
	At Screening	21 Days after 2nd Vaccination	21 Days after 3rd Vaccination
**Total participants**			
Number of participants, N	199	175	168
Seroconversion rate (%, 95% CI)	–	80 (45.7%, 38.5–53.1%)	122 (72.6%, 65.4–78.8%)
GMT (95% CI)	15.97 (11.52–22.14)	86.60 (65.03–115.33)	145.76 (114.10–186.21)
GMT fold increase relative to screening	–	4.76	7.88
**Seropositive at screening**			
Number of participants, N	131	120	116
Seroconversion rate (%, 95% CI)	–	35 (29.2%, 21.8–37.8%)	76 (65.5%, 56.5–73.5%)
GMT (95% CI)	68.72 (53.22–88.74)	197.15 (152.26–255.27)	291.22 (239.58–353.98)
GMT fold increase relative to screening	–	2.87	4.26
**Seronegative at screening**			
Number of participants, N	68	55	52
Seroconversion rate (%, 95% CI)		45 (81.8%, 69.7–89.8%)	46 (88.5%, 77–94.6%)
GMT		31.13 (20.21–47.94)	14.39 (9.31–22.24)

GMT—geometric mean titer.

**Table 7 viruses-15-01828-t007:** Seroconversion rates (in NT) of randomized participants of phase II and IIb.

Group	Positive (%, 95% CI)		
	II (18–60 years)	IIb (Age 60+ years)	
	28 d Post 2nd Vaccination	21 d Post 2nd Vaccination	21 d Post 3rd Vaccination
**All participants**			
Vaccine	N = 122	N = 175	N = 168
	79 (64.8%, 55.9–72.7%)	80 (45.7%, 38.5–53.1%)	122 (72.6%, 65.4–78.8%)
Placebo	N = 45	–	–
	2 (4.5%, 1.3–15.1)		
**Participants Seronegative at Screening**			
Vaccine	N = 69	N = 55	N = 52
	60 (86.9%, 77.0–93%)	45 (81.8%, 69.7–89.8%)	46 (88.5%, 77.0–94.6%)
Placebo	N = 27	–	–
	1 (3.7%, 0.7–18.3%)		
**Participants seropositive at screening**			
Vaccine	N = 53	N = 120	N = 116
	19 (35.8%, 24.3–49.3%)	35 (29.2%, 21.8–37.8%)	76 (65.5%, 56.5–73.5%)
Placebo	N = 17	–	–
	1 (5.9%, 1.0–27.0%)		

## Data Availability

The data presented in this study are available on request from the corresponding author. The data are not publicly available due to the privacy of study participants.

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
