# Peer review of "Safety and Immunogenicity of Inactivated Whole Virion COVID-19 Vaccine CoviVac in Clinical Trials in 18–60 and 60+ Age Cohorts"

_viruses, 2023, doi:10.3390/v15091828_

Round 1

Reviewer 1 Report

This is an excellent paper reporting safety and immunogenicity data of an attenuated SARS-CoV-2 vaccine in a clinical study. The study is still very timely because there is a need for alternative vaccines besides genetic vaccines especially because  vaccines are needed which can be used safely for boosting SARS-CoV-2-specific antibody responses. I suggest that the authors mention in the introduction the limitations of genetic vaccines especially regarding side effects and utility for boosting. I further suggest that they relate their findings regarding safety of their vaccine with reported safety findings for genetic vaccines as far as this is possible considering the limited number of patients treated in the current paper and safety data obtained for larger numbers of subjects for genetic vaccines in the discussion. The authors should also mention if their technology can be used for different SARS-CoV-2 variants, in particular Omicron and eventually upcoming new variants. Furthermore, their finding that neutralizing antibody responses can be only boosted when they have declined to a certain level is is extremely interesting and should be highlighted more. Their data would support the use of tests for measuring blocking antibodies as described (doi:10.3390/ijms24065352) to identify subjects who may respond best to booster vaccination. In line 164 the authors mention that participants of stage 3 were included regardless their SARS-CoV-2 antibody status at baseline. I therefore think that it is not appropriate to speak about seroconversion in Table 5 for the placebo group because at screening 37.8% were already positive and there was not much of an increase (i.e., 40.9%) 28 days after the 2nd vaccination. The same is true for the vaccinated group. Perhaps Table 5 and the accompanying text could be presented to take this into consideration.

Author Response

This is an excellent paper reporting safety and immunogenicity data of an attenuated SARS-CoV-2 vaccine in a clinical study. The study is still very timely because there is a need for alternative vaccines besides genetic vaccines especially because vaccines are needed which can be used safely for boosting SARS-CoV-2-specific antibody responses.

Response: Dear Reviewer, thank you very much for the thorough review and appreciation of our manuscript. The answers to the specific comments are provided below.

>I suggest that the authors mention in the introduction the limitations of genetic vaccines especially regarding side effects and utility for boosting. I further suggest that they relate their findings regarding safety of their vaccine with reported safety findings for genetic vaccines as far as this is possible considering the limited number of patients treated in the current paper and safety data obtained for larger numbers of subjects for genetic vaccines in the discussion.

Response: Introduction was amended with statements and references on genetic vaccines side effects and their limitations for use in routine immunization. Some additional conclusions on safety were included into Discussion.

>The authors should also mention if their technology can be used for different SARS-CoV-2 variants, in particular Omicron and eventually upcoming new variants.

Response: Thank you for the comment. CoviVac production platform uses an infectious virus primarily isolated from the natural viral population multiplied in Vero cells expressing ACE2 receptor (L. I. Kozlovskaya et al., 2021). Therefore, this platform can be easily modified to use other SARS-CoV-2 variant as vaccine antigen. Such modified vaccines can be used as heterologous booster in the ever-changing epidemiology of COVID-19.

The text above was added to the Discussion section.

>Furthermore, their finding that neutralizing antibody responses can be only boosted when they have declined to a certain level is extremely interesting and should be highlighted more. Their data would support the use of tests for measuring blocking antibodies as described (doi:10.3390/ijms24065352) to identify subjects who may respond best to booster vaccination.

Response: Thank you for the comment. Indeed, subjects with preexisting nAT in high titers cannot respond properly to a booster immunization, and in these cases booster immunization can be inefficient. Therefore, testing of pre-existing immunity may be required before booster vaccination to understand its necessity.

The text above was added to the Discussion section.

>In line 164 the authors mention that participants of stage 3 were included regardless their SARS-CoV-2 antibody status at baseline. I therefore think that it is not appropriate to speak about seroconversion in Table 5 for the placebo group because at screening 37.8% were already positive and there was not much of an increase (i.e., 40.9%) 28 days after the 2nd vaccination. The same is true for the vaccinated group. Perhaps Table 5 and the accompanying text could be presented to take this into consideration.

Response: Thank you for the comment. We do agree with this statement from scientific point of view. However, seroconversion rates (defined as 4-fold or higher increase in GMT of specific antibodies from baseline in the NT) were the outcomes of the trial, so they had to be calculated and presented.

Reviewer 2 Report

In the manuscript, the authors Gordeychuk&Kozlovskaya et al provide data safety data from a clinical phase I/II and IIb trial regarding an inactivated whole virion SARS-CoV-2 vaccine. The design of the study is clearly presented, the research design is appropriate and the methods are adequately described. Importantly, the SARS-CoV-2 vaccine named CoviVac led to no significant adverse events as recorded by the study centers and described by the authors.

However, there are some points that should get discussed:

- The authors could include in the introduction also other approaches and novel strategies for SARS-CoV-2 vaccines and treatment.

- In the phase I/II study (18-60 years) and the phase II study (years 60+) different vaccination regimen were used. It would be important to know why different intervals and number injections for the primary immunization were chosen. It has been shown for other SARS-CoV-2 vaccines (homologous mRNA or heterologous mRNA/vector) that a third booster immunization is necessary for the induction of a sustained neutralizing antibody response. The authors may also discuss why they haven´t included a third immunization in the phase I/II study (18-60 years).

- Immunogenicity assessment of CoviVac was performed by measuring seroconversion rates of neutralizing antibodies and fold increase to baseline neutralizing antibody titers. In the discussion the authors mention that CoviVac can successfully be used for primary immunization and booster. But the authors missed to discuss this outcome in the context of current SARS-CoV-2 omicron variants. Other studies have shown that nABs to XXB variants are 100 folder lower than to the ancestral virus strain (doi: 10.1093/infdis/jiad111). What are the nAB titers induced by CoviVac to current omicron variants?

- Is there is a defined cut-off of virus neutralizing titer and protection of infection?

Minor

- It should be mentioned clearly in the methods section which SARS-CoV-2 variant (alpha?) was used for the vaccine and the neutralization tests.

- No results of CMIA assay are shown are there is no rationale between nABs and Nucleocapsid protein  IgG, can be deleted from line 261-262 and 449-451.

- Tables 3 and 4 are hard to read when separated on different pages, one might try to have one table complete on one page

- Table 5 can be improved by explaining the numbers in braces

- Line 566 should read: the third vaccination

- Line 620 should read schedule of two injections

Language editing is required: many errors with capital letters

Author Response

In the manuscript, the authors Gordeychuk&Kozlovskaya et al provide data safety data from a clinical phase I/II and IIb trial regarding an inactivated whole virion SARS-CoV-2 vaccine. The design of the study is clearly presented, the research design is appropriate and the methods are adequately described. Importantly, the SARS-CoV-2 vaccine named CoviVac led to no significant adverse events as recorded by the study centers and described by the authors.

Response: Dear Reviewer, thank you very much for the thorough review and appreciation of our manuscript. The provided comments were highly valuable to improve the discussion section. The answers to the specific comments are provided below.

>However, there are some points that should get discussed:

- The authors could include in the introduction also other approaches and novel strategies for SARS-CoV-2 vaccines and treatment.

Response: Thank you for the comment. The text about currently used treatment regimens was added to the Introduction section.

>In the phase I/II study (18-60 years) and the phase II study (years 60+) different vaccination regimen were used. It would be important to know why different intervals and number injections for the primary immunization were chosen. It has been shown for other SARS-CoV-2 vaccines (homologous mRNA or heterologous mRNA/vector) that a third booster immunization is necessary for the induction of a sustained neutralizing antibody response. The authors may also discuss why they haven´t included a third immunization in the phase I/II study (18-60 years).

Response: Thank you for the comment. Study participants from different age cohorts were vaccinated according to different schedules: phase II participants (aged 18-60) received 2 doses of the vaccine with 14 days interval whereas phase IIb participants (aged 60+) received 3 doses with 21 days intervals. Changes in vaccination schedules were made to provide the most favorable conditions of anti-SARS-CoV-2 immunity development in 60+ recipients, who are expected to have declined immune response. The results of the phase II showed 81.8% seroconversion rate 21 days after the 2nd vaccination (in seronegative vaccinees). To achieve the same rates in 60+ cohort we increased the time between vaccinations and added 3rd vaccination. This allowed us to achieve seroconversion in 88.5% of participants 21 days after the 3rd vaccination in phase IIb. It is obvious that an increase in time between vaccinations and additional dose of the vaccine would favor the phase II participants as well. However, CoviVac development and phase II were conducted in 2020 during the rise of COVID-19 pandemic. The vaccine providing anti-SARS-CoV-2 immunity in a short period of time was extremely needed. Therefore, a shortest appropriate schedule was chosen for phase II and following use. Nevertheless, in the present time 3 vaccinations schedule with 21 days between injections seems appropriate.

The text above was added to the Discussion section.

>- Immunogenicity assessment of CoviVac was performed by measuring seroconversion rates of neutralizing antibodies and fold increase to baseline neutralizing antibody titers. In the discussion the authors mention that CoviVac can successfully be used for primary immunization and booster. But the authors missed to discuss this outcome in the context of current SARS-CoV-2 omicron variants. Other studies have shown that nABs to XXB variants are 100 folder lower than to the ancestral virus strain (doi: 10.1093/infdis/jiad111). What are the nAB titers induced by CoviVac to current omicron variants?

Response: Thank you for the comment. Some serum samples collected during phase II were tested against strains of later circulating SARS-CoV-2 variants: Delta and Omicron (BA.2). Overall 57.1% and 61.9% samples neutralized Delta and Omicron strains (L. Kozlovskaya et al., 2022), respectively. These results are in line with the outcomes of previous experiments confirming that CoviVac contains a whole virion with trimeric intact spikes (Bagrov et al., 2022), thus vaccination with CoviVac induce a wide range of antibodies, including to cross-neutralizing epitopes of the S protein. We expect further decline in the immunity against novel Omicron variants. However, boosting with vaccines based on the novel variants can be a solution, maintaining neutralizing antibodies developed as a response to a primary CoviVac immunization. CoviVac production platform uses an infectious virus primarily isolated from the natural viral population multiplied in Vero cells expressing ACE2 receptor (L. I. Kozlovskaya et al., 2021). Therefore, this platform can be easily modified to use other SARS-CoV-2 variant as vaccine antigen. Such modified vaccines can be used as heterologous booster in the ever-changing epidemiology of COVID-19.

The text above was added to the Discussion section.

>- Is there is a defined cut-off of virus neutralizing titer and protection of infection?

Response: Thank you for the comment. Generally, low levels of neutralizing antibodies (i.e. below 1:20) are considered to be protective against severe course of COVID-19, but unfortunately there is no defined correlate of protection against infection.

Minor

>- It should be mentioned clearly in the methods section which SARS-CoV-2 variant (alpha?) was used for the vaccine and the neutralization tests.

Response: Materials and Methods section ‘Immunogenicity assessment’ includes genetic description of NT and vaccine strains (Lines 277-281). Both strains belong to prototype B.1.1 lineage, so the main difference from Wuhan virus is D614G mutation in S protein.

>- No results of CMIA assay are shown are there is no rationale between nABs and Nucleocapsid protein IgG, can be deleted from line 261-262 and 449-451.

Response: Thank you for the comment. Indeed, the CMIA results poorly correlated with NT. Still, we believe that the data provided in Supplementary Table S5, Supplementary Figure S1 might be valuable to the specialists working on sensitivity assessment of serological test. The text from lines 261-262 was removed

>- Tables 3 and 4 are hard to read when separated on different pages, one might try to have one table complete on one page

Response: Thank you for the comment. We agree that the large tables are hard to read when separated and will try to modify this in the final version before publication.

>- Table 5 can be improved by explaining the numbers in braces

Response: Thank you for the comment. The numbers in braces were incompletely explained in the upper row of the table. The text was corrected as “Positive, n (%, 95% CI)”

>- Line 566 should read: the third vaccination

Response: Thank you for the comment. The text was corrected according to the Reviewer’s recommendation.

>- Line 620 should read schedule of two injections

Response: Thank you for the comment. The text was corrected according to the Reviewer’s recommendation.

Reviewer 3 Report

This study presents interesting data on safety and immunogenicity of inactivated whole virion vaccine. The main research findings of this paper will be important for public health and be able to increase vaccine potions. I have just a few comments, mainly on writing style, which should help to improve the clarity of the contribution.

Specific comments:

1.          Line 264: There appears to be a lack of method of serum separation. I would anticipate that adding this more information.

2.          Line 340-344 and Table2: The authors should explain that the demographic data of vaccine group and placebo group are not significantly difference.

3.          Line 425-427: It is difficult to understand whether there were serious AEs or not. This should be better described and clarified.

4.          Table 5: The participants in Table 5 did not included seropositive at screening (Day 0). But 53 in vaccine group and 17 of placebo group were showed positive for NT. The authors should explain why that groups showed seropositive.

5.          Line 483: When is the value of the timepoint?

6.          Line 510-511: Is this the data of 60+ age group? It would be useful to include this information for better understanding.

7.          Line 537-545: This was already explained in Introduction. It is better to summarize according to the contents described in the study outcomes.

8.          Line 604-610 and Table 7: I think this sentence is the results. Please move to results section. In addition, the results of Table 6 and Table 7 was overlapped. Table 7 is better to include into Table 6.

9.          Line 610-612: Explanation for the relationship between age, seroconversion rate and duration of antibody titer should be provided using references.

10.          Line 629-632: How about mentioning whether other inactivated whole virion vaccines can also increase IFN-γ levels?

11.          There are a number of typographical errors throughout the manuscript. If you have performed third-party English proofreading, please indicate so. If not, please provide proofreading.

12.          The terms are different in each section.

- In “Study design and participants” section from Line 149, authors were used as “participants”. But authors were used as “persons” in “discussion”.

- Line 592 and 599: I think it is “vaccination”, not “immunization”.

- Line 599, 614 and 618-619: it is “phase IIb” not ”stage IIb”.

I hope these comments will be helpful.

There are a number of typographical errors throughout the manuscript. If you have performed third-party English proofreading, please indicate so. If not, please provide proofreading.

Author Response

This study presents interesting data on safety and immunogenicity of inactivated whole virion vaccine. The main research findings of this paper will be important for public health and be able to increase vaccine potions. I have just a few comments, mainly on writing style, which should help to improve the clarity of the contribution.

Response: Dear Reviewer, thank you very much for the thorough review and appreciation of our manuscript. The provided comments allowed us to increase the readability of the manuscript and to unify the presentation of the results of phase I/II and phase IIb trials. The answers to the specific comments are provided below.

Specific comments:

  1. Line 264: There appears to be a lack of method of serum separation. I would anticipate that adding this more information.

Response: For serum separation, blood was collected into Vacuette tubes with clot activator and serum separator gel. Serum was separated via centrifugation, aliquoted into 1.7 ml tubes and stored frozen for further analysis for anti-SARS-CoV-2 antibodies.

This text was added to the Materials and Methods section.

  1. Line 340-344 and Table2: The authors should explain that the demographic data of vaccine group and placebo group are not significantly difference.

Response: The differences in demographic and anthropometric characteristics in Table 1 are statistically insignificant. The statement in the Materials and Methods section was amended.

Demographic distribution in 60+ age group corresponds with the one in Russian population i.e. the target population for vaccination.

The results between 18-60 and 60+ age groups are not being compared with each other, but presented collectively. Thus, the differences in sex distribution between these 2 cohorts are not important.

  1. Line 425-427: It is difficult to understand whether there were serious AEs or not. This should be better described and clarified.

Response: Thank you for the comment. The sentence was rewritten, as the previous version was unclear.

  1. Table 5: The participants in Table 5 did not included seropositive at screening (Day 0). But 53 in vaccine group and 17 of placebo group were showed positive for NT. The authors should explain why that groups showed seropositive.

Response: Thank you for the comment. As stated in the “Study design and participants” section, only the participants of Stages 1 and 2 in the phase I/II trials were included based of serological tests, while the participants of Stage 3 (immunogenicity assessment) were included regardless of their SARS-CoV-2 IgM and IgG status at enrolment, thus the data presented in Table 5 are correct. As mentioned by the Reviewer, 53 participants in vaccine group and 17 of placebo group were seropositive in NT at screening.

  1. Line 483: When is the value of the timepoint?

Response: Thank you for the comment. That was at the end of the study, i.e. day 28 after the second vaccination. The clarification was added to the manuscript.

  1. Line 510-511: Is this the data of 60+ age group? It would be useful to include this information for better understanding.

Response: Thank you for the comment. The title of the table was corrected

  1. Line 537-545: This was already explained in Introduction. It is better to summarize according to the contents described in the study outcomes.

Response: The text was corrected according to the Reviewer’s recommendations

  1. Line 604-610 and Table 7: I think this sentence is the results. Please move to results section. In addition, the results of Table 6 and Table 7 was overlapped. Table 7 is better to include into Table 6.

Response: Thank you for the comment. Table 5 and Figure 3 present the seroconversion rates and GMT in phase II study in 18–60 cohort, including the impact of baseline antibody levels on seroconversion rates. Table 6 presents seroconversion rates and nAB levels in 60+ cohort in phase IIb. We believe that Table 7 presented in Discussion summarizing the data presented in Table 5, Figure 3 and Table 6 helps to compare the data obtained in phases II and IIb. Clarifications were added to the figure and table titles.

  1. Line 610-612: Explanation for the relationship between age, seroconversion rate and duration of antibody titer should be provided using references.
    Response: Thank you for the comment. On the contrary, in our study seroconversion rates in 18-60 and 60+ cohorts were similar. Additional increase of seroconversion rate in participants seropositive at screening was observed in 60+ cohort only after the third dose. The text in lines 610-612 was corrected.

  1. Line 629-632: How about mentioning whether other inactivated whole virion vaccines can also increase IFN-γ levels?

Response: Thank you for the comment. S protein-specific IFN-γ production by T-cells of individuals after two doses of inactivated whole-virion adjuvanted COVID-19 vaccine was previously reported by Lazarus et al. Text and reference were added to the manuscript.

  1. There are a number of typographical errors throughout the manuscript. If you have performed third-party English proofreading, please indicate so. If not, please provide proofreading.

Response: Thank you for the recommendation. The text will be processed by an English editing service in the final version before publication.

  1. The terms are different in each section.

- In “Study design and participants” section from Line 149, authors were used as “participants”. But authors were used as “persons” in “discussion”.

Response: The text was corrected according to the Reviewer’s recommendation.

- Line 592 and 599: I think it is “vaccination”, not “immunization”.

Response: The text was corrected according to the Reviewer’s recommendation.

- Line 599, 614 and 618-619: it is “phase IIb” not ”stage IIb”.

Response: The text was corrected according to the Reviewer’s recommendation.

Round 2

Reviewer 2 Report

The comments and corrections provided be the authors highly improved the manuscript and it is now suitable for publication.

However, Figure 4 has a low quality and should be substituted by a high quality figure and line 770 nAT should read nAB